# NADPH Oxidases: The Vital Performers and Center Hubs during Plant Growth and Signaling

**DOI:** 10.3390/cells9020437

**Published:** 2020-02-13

**Authors:** Chun-Hong Hu, Peng-Qi Wang, Peng-Peng Zhang, Xiu-Min Nie, Bin-Bin Li, Li Tai, Wen-Ting Liu, Wen-Qiang Li, Kun-Ming Chen

**Affiliations:** 1State Key Laboratory of Crop Stress Biology in Arid Areas, College of Life Sciences, Northwest A&F University, Yangling 712100, Shaanxi, China; 2College of Life Science and Agronomy, Zhoukou Normal University, Zhoukou 466000, Henan, China; 3School of Life Sciences, Tsinghua University, Beijing 100084, China

**Keywords:** NADPH oxidases (NOXs/RBOHs), reactive oxygen species (ROS), activity regulation, plant signaling, crop improvement

## Abstract

NADPH oxidases (NOXs), mostly known as respiratory burst oxidase homologs (RBOHs), are the key producers of reactive oxygen species (ROS) in plants. A lot of literature has addressed ROS signaling in plant development regulation and stress responses as well as on the enzyme’s structure, evolution, function, regulation and associated mechanisms, manifesting the role of NOXs/RBOHs as the vital performers and center hubs during plant growth and signaling. This review focuses on recent advances of NOXs/RBOHs on cell growth, hormone interaction, calcium signaling, abiotic stress responses, and immunity. Several primary particles, including Ca^2+^, CDPKs, BIK1, ROPs/RACs, CERK, FER, ANX, SnRK and SIK1-mediated regulatory mechanisms, are fully summarized to illustrate the signaling behavior of NOXs/RBOHs and their sophisticated and dexterous crosstalks. Diverse expression and activation regulation models endow NOXs/RBOHs powerful and versatile functions in plants to maintain innate immune homeostasis and development integrity. NOXs/RBOHs and their related regulatory items are the ideal targets for crop improvement in both yield and quality during agricultural practices.

## 1. Introduction

Reactive oxygen species (ROS) such as superoxide anion (·O_2_^−^) and hydrogen peroxide (H_2_O_2_) are known to participate in regulation of a wide range of biological processes involved in plant development and growth, as well as adaptation to biotic and abiotic stresses [1,2,3,4,5]. NADPH oxidases (NOXs), also well known as respiratory burst oxidase homologs (RBOHs), are key enzymes for ROS production in plants. As the molecular “hubs” during ROS-mediated signal transduction pathways, NOXs/RBOHs in plants have recently received considerable attention, and more and more NOX/RBOH gene homologs have been identified in a variety of plant species [6,7,8]. The functions and regulatory mechanisms of the NOXs/RBOHs in plants have been successively reported. Recently, we identified 46 NOX family genes in wheat genome, and a systematic analysis in their transcriptional expressions reveals their vital but diverse roles in plant growth regulation and stress responses [8]. To date, more than 150 protein members of the NOXs/RBOHs family have been identified and/or characterized in various plant species (Appendix A) and among them, at least 56 members have been fully elucidated in functions. These NOXs are expressed in whole plants or specific organs/tissues relying on development stage and are functioning in many developmental processes and stress responses (Appendix A). In addition, recent advances in plant cell growth, hormone interaction and calcium signaling, as well as biotic and abiotic stress responses, have shown that NOXs/RBOHs may act as the center hubs of multiple signaling pathways of plants. Many important signaling systems functioning in plant immunity and/or hormonal and abiotic stress responses, including, OsCERK1/OsCEBiP(OsLyp4/6)-OsRLCK176/185, FLS2(EFR)/BAK1-BIK1, CERK1/LYK5(LYM1/3)-BIK1, FER context and hormonal signaling stuffs, are all directly or indirectly associated with NOXs/RBOHs activation regulation. In this review, we focus on the major advances of NOXs/RBOHs on their structure, biological function, activity regulation and signaling pathways involved in plant growth regulation and stress responses. Based on these results, we summarized the major signaling models involved in regulation of NOXs/RBOHs activity and maintenance of intracellular ROS balance in plants, hoping to provide valuable information for further studies in this field and important cues for crop improvement by bioengineering and molecular breeding during agricultural practices.

## 2. Protein Structure and Evolution of Plant NOXs/RBOHs

### 2.1. Highly Conservative Structure

NADPH oxidases (NOXs) are key enzymes of ROS generation and thus play crucial roles in a variety of biological processes in different kingdoms of life [9,10,11]. Since the first NADPH oxidase was identified in human phagocytic cells, more and more NOXs/RBOHs were identified in many other species including animals, higher plants and fungi [5,7,8]. In fungi, there are four types of NOXs assigned as NOXA, NOXB, NOXC and FRE [12]. In animals, there are seven types of NOXs, named as NOX1, NOX2, NOX3, NOX4, NOX5, DUOX1 and DUOX2 [11]. However, in plants, only NOX5-like-type NOXs were found except the ancestral NOXs, ferric reduction oxidases (FROs), which are considered as the isoforms of yeast ferric-chelate reductases (FREs) [6,7,11]. All NOX family members are transmembrane proteins that transport electrons across biological membranes to reduce oxygen to superoxide anion (·O_2_^−^) [10]. In accordance with this preserved function, there are conserved structural properties of NOX enzymes that are common to all family members.

The earliest and most comprehensive studies about NOXs were largely carried out in phagocytic cells, in which their functions involve the destruction of pathogenic microorganisms by producing ROS [13]. The phagocytic cell, NADPH oxidase, is a protein complex that is comprised of several protein components, which can be divided into two groups. The first is a membrane-bound NADPH-binding flavocytochrome *b*_558_, which is comprised of a glycosylated transmembrane protein gp91^phox^ (β subunit) and a non-glycosylated p22^phox^ subunit (α subunit). The gp91^phox^ contains the entire electron transport chain from NADPH to molecular oxygen to produce ·O_2_^−^ outside the plasma membrane [14,15]. The second group comprises four regulatory proteins: p47^phox^, p67^phox^ and p40^phox^ phosphoproteins and Rac1 or Rac2 (small GTP (Guanosine triphosphate)-binding protein). These four regulatory proteins could be translocated to the plasma membrane and form an enzyme complex with the first group after being activated. In this process, Rac1 or Rac2 is independently activated and translocated to assemble with the oxidase effector system, each of them endows the enzyme with guanine nucleotide sensitivity as it interacts with the activator component, p67^phox^, in a GTP-dependent manner. The p47^phox^ component acts as a critical phosphorylation-dependent adaptor molecule that enables the interactions between p67^phox^ and the flavocytochrome [16]. Further studies have shown that NADPH oxidase is also present in many other types of mammalian cells where the level of ROS generated by it is much lower and the processes of ROS generation last much longer. In this case, ROS can be used as regulatory and signal molecules for modulation of metabolic processes, leading to various biological effects [17].

Although NOXs have many isoforms (homologs), the full spectrum is only found in the animal kingdoms. A mammal genome generally contains seven genes encoding gp91^phox^ homologs: five close relatives of gp91^phox^ homologs (Nox1–Nox5), and two distant relatives of gp91^phox^ homologs (Duox1 and Duox2) [10,18]. All of them share FAD (flavin adenine dinucleotide) - and NADPH-binding sites in the C-terminal domain, two heme binding sites, a functional oxidase domain responsible for ·O_2_^−^ production, and some degree of sequence similarity. However, Nox5, Duox1 and Duox2 are different kinds relative to Nox1-4: Nox5 possesses four sites (motifs) of the EF hand type (elongation factor), while both Duox1 and Duox2 possess two sites of the EF hand type, which are characteristic of specific calcium-binding domains [17]. Moreover, except for Duox1 and Duox2 which possess an extra transmembrane domain with peroxidase activity, all of these homologs contain six transmembrane domains.

In stark contrast to the multienzyme complex of NADPH oxidase in mammals, plants only possess NOX5-like NOXs, which are also called respiratory burst oxidase homologs (RBOHs), even though multiple members exist in different species [6,11,19]. The typical NOXs all possess four conserved domains, namely NADPH_Ox (Pfam accession number PF08414), Ferric_reduct (PF01794), FAD_binding_8 (PF08022) and NAD_binding_6 domain (PF08022). At the same time, the distribution of amino acid residues in every domain is quite similar but not identical among the NOX/RBOH members [8]. However, further studies indicate some exceptions. The cytoplasm of tomato suspension culture cells contains protein components similar to animal p67phox and p40phox, which are activated by a fungal elicitor and can move to the plasmalemma and incorporate into the membrane cytoskeleton [17,20]. Furthermore, our recent research showed that some members have one more NADPH_Ox domain, such as TaNOX3 and TaNOX6 possessing two NADPH_Ox domains in wheat [8]. Besides the typical NOXs, some ferric reduction oxidases (FROs), which are considered as the isoforms of yeast FREs, were also found in high plants [6]. It was identified that FROs are closely related to typical plant NOXs but differences still exist: FROs contain six membrane-spanning domains, two hemes and conserved motifs involved in NADPH and FAD binding but lack NADPH_Ox domain and several calcium-binding EF-hand motifs that the typical NOX proteins possess [10,19,21]. Moreover, another new type of NOXs was put forward and assigned as NOX-likes, which have an NADPH_Ox domain but lack one or two other domains that typical NOXs possess [8]. 

### 2.2. Complex Evolution History

The typical NOX family, being identified only in terrestrial plants, underwent a complex evolution. According to our previous study, NOXs and FROs in plants can be divided into four well-conserved groups represented as NOX, FRO I, FRO II and FRO III, and an evolution model was constructed [7]. In this model, all FRO family members originated from a common ancestor which contains only the Ferric_reduct domain. During the evolutionary process, this ancestor obtained FAD_binding_8 and NAD_binding_6 domains first by gene fusion and duplication, and then clustered into FRO I, FRO II and FRO III subfamilies. FRO III mainly exists in fungi and FRO I, perceived as ancient NOX, mainly exists in animals and in two kinds of algae (rhodophytes and chlorophytes). After that, FRO I obtained another important domain-NADPH_Ox and converted into the typical NOXs in plants. FRO IIs exist both in chlorophytes and land plants but NOXs only exist in land plants. Clearly, the gene constructions become more and more complicated from FROs to NOXs over the course of evolution. In addition to domain gain and gene duplication, gene fusion as well as exon shuffling might also be involved in the evolution for biological diversity and functional divergence of the NOX gene family [7,22,23,24,25]. More recently, we found that TaFROs encoded by the genes located on Chr 1 are much closer to the typical TaNOXs in phylogenetics than those on Chr 2, and the TaFROs on Chr 2 might be more ancient forms of TaNOXs in wheat [8].

## 3. NOXs/RBOHs in Plant Development

The wide existence of NOXs/RBOHs in plants in some way indicates their great importance. Experiments from recent studies did show that NOXs/RBOHs play important and diverse roles in many areas including development regulation as well as the response to biotic and abiotic stresses and hormone signaling (Figure 1).

### 3.1. Pollen Germination and Pollen Tube Growth

NOX enzyme, by producing ROS, is needed to sustain the normal rate of pollen tube growth and this is likely to be a general mechanism in the control of tip growth of polarized plant cells [26,27]. NOXs generate tip-localized, pulsating amounts of H_2_O_2_ that functions, possibly through Ca^2+^ channel activation, to maintain a steady, tip-focused Ca^2+^ gradient in pollen tube tip during growth [28]. On the other hand, pollen NOX can be activated by Ca^2+^ and Ca^2+^ can increase its activity in vivo [26,29]. This process would happen especially when the pollen grains are hydrated under mild conditions, so the activity of pollen NOX could be concentrated in those insoluble fractions, which could facilitate the exposure of tissues to ROS produced by this enzyme. It is worth mentioning that the extent of this exposure could differ among the plant families according to where the NOX resides in pollen grains and during hydration in the mucosa [30]. In addition, pollen NOX can also be activated by low abundant signaling phospholipids, such as phosphatidic acid (PA) and phosphatidylinositol 4, 5-bisphosphate in vitro and in vivo, so there is a possible synergism between Ca^2+^ and phospholipid-mediated NOX activation in pollen [29]. In plants, ROP/RAC GTPases (the subfamily of Rho-type GTPases, which belong to the Rat sarcoma superfamily of small GTP-binding proteins) are also necessary for normal pollen tube growth by regulating ROS production [29]. Besides these items mentioned above, O_3_ was also indicated to increase ragweed pollen allergenicity through stimulation of ROS-generating NADPH oxidase [31]. Recently, two NOXs/RBOHs, AtRbohH and AtRbohJ, have been shown to localize at the plasma membrane of pollen tube tip and the ROS production by the NOXs/RBOHs presumably plays a critical role in the positive feedback regulation that maintains the pollen tube tip growth [32,33]. Furthermore, apoplastic ROS derived from AtRbohH and AtRbohJ are involved in pollen tube elongation by affecting the cell wall metabolism [33]. More intriguing is that ectopic expressing of AtRbohC/hair root2 (hrd2) in pollen tubes could restore *atrbohH* and *atrbohJ* defects in tip growth of pollen tubes [34], which implies that AtRbohC/hrd2 also plays roles in regulating the development of pollen tubes. Moreover, AtRbohE was also suggested to be critical for programmed cell death (PCD) and pollen development in *Arabidopsis thaliana* L. [35]. Meanwhile, OeRbohH, possessing a high degree of identity with AtRbohH/J, plays an important role in pollen germination and pollen tube growth in olive [36]. Genetic interference with the temporal ROS pattern by manipulating *NOX/RBOH* genes, affected the timing of tapetal PCD (programmed cell death) and resulted in aborted male gametophytes [35]. All in all, we still cannot figure out how many factors are related to the regulation of pollen NOX. Nevertheless, what we already know is that pollen NOXs play a significant part in the regulation of pollen germination.

### 3.2. Root and Root Hair Development 

The elongation of roots and root hairs is essential for uptake of minerals and water from the soil. Ca^2+^ influx from the extracellular store is required for cell elongation in roots [37]. It was suggested that plasma membrane NOXs/RBOHs and H^+^-ATPases (a H^+^ pump by coupling with energy of ATP hydrolysis on plasma membranes) are functionally synchronized and they work cooperatively to maintain the membrane electrical balance while mediating plant cell growth through wall relaxation [38]. Observations on maize roots indicate that the activities of plasma membrane-associated NADPH oxidase respond both to gravity and to imposed centrifugal forces [39]. In an early study, AtRHD2, a NADPH oxidase in *Arabidopsis*, was reported as controlling root development by making ROS that regulates plant cell expansion through the activation of Ca^2+^ channels [40]. Further, both AtRbohC/RHD2 and ROP (RHO of plants) GTPases were found to be required for normal root hair growth by regulating ROS production [41]. Coincidentally, the maize (*Zea mays* L.) *roothairless5* (*rth5*) which encodes a monocot-specific NADPH oxidase, was found to be responsible for establishing the high levels of ROS in the tips of growing root hairs [42]. In rice, *OsNOX3* was also reported to play critical roles in root hair initiation and elongation by regulating the content of superoxide and hydrogen peroxide in root hair tips [27]. In addition, both AtRbohD and AtRbohF are essential for ABA (abscisic acid)-promoted ROS production in *Arabidopsis* roots, and ROS subsequently activate Ca^2+^ signaling as well as reduce auxin sensitivity of roots, thus positively regulating ABA-inhibited primary root growth [43]. Moreover, AtRbohD and AtRbohF negatively modulate lateral root development by changing the peroxidase activity and increasing the local generation of ·O_2_^-^ in primary roots in an auxin-independent manner [44]. Similar results were also acquired in the legume-rhizobia symbiosis and legumes use different RBOHs for different stages of nodulation [45,46]. Moreover, nitric oxide (NO) can activate NADPH oxidase activity, resulting in increased generation of ·O_2_^-^, which subsequently induces growth of adventitious roots and acts downstream of auxin action in the process of root growth and development [47]. These results suggest a vital role of NOXs/RBOHs in root and root hair development in plants.

### 3.3. Seed Germination

NOXs/RBOHs also play important roles in seed germination. The functional mechanism has been proposed in many plant species, such as *Arabidopsis thaliana* L. rice (*Oryza sativa* L.) and barley (*Hordeum vulgare* L.). AtRbohB is a major producer of ·O_2_^-^ in germinating seeds, and inhibition of the ·O_2_^-^ production by diphenylene iodonium (DPI) leads to a delay in seed germination of *Arabidopsis* and cress [48]. In rice, OsNOX5, 7 and 9 might play crucial roles in radicle and root elongation during seed germination by regulating ROS production [49]. Similarly, ·O_2_^-^ produced by NADPH oxidase also regulates seed germination and seedling growth in barley [50,51]. Moreover, NOX/RBOH-mediated ROS production promotes gibberellic acid (GA) biosynthesis in barley embryos through regulation of HvKAO1 and HvGA3ox1 proteins, while GA induces and activates NOXs/RBOHs for ROS production in aleurone cells to induce α-amylase activity of the cells and therefore increases seed germination [5,52,53].

### 3.4. Plant-Microorganism Ineractions

*Phaseolus vulgaris* NADPH oxidase is crucial for successful rhizobial colonization and probably maintains proper infection thread growth and shape [54]. Moreover, it also has critical roles in reducing arbuscular mycorrhizal fungal (AMF) colonization. Overexpression of PvRbohB augments nodule efficiency by enhancing nitrogen fixation and delaying nodule senescence but impairs AMF colonization [55]. A *Medicago truncatula* NADPH oxidase, MtRbohA, has similar effects. It is significantly upregulated in *Sinorhizobium meliloti*-induced symbiotic nodules, while hypoxia prevailing in the nodule-fixing zone may stimulate MtRbohA expression, which would, in turn, lead to the regulation of nodule functioning [56]. Moreover, NoxA, a NADPH oxidase isoform in the grass endosymbiont *Epichloë festucae*, was identified as essential for the establishment of systemic compatible infections in host plants [50]. ROS produced by NoxA or NoxB (from *Fusarium oxysporum*) regulate hyphal growth of a fungal pathogen towards roots of the host plants to maintain a mutualistic and symbiotic interaction [57,58]. Recently, four GmNOXs from soybean genome (*Glycine max*) also showed strong expression in nodules, pointing to their probable involvement in nodulation [59]. All these results suggest that, NOXs/RBOHs, regulated by several different factors, play a significant role in mutualistic and symbiotic processes between plant and microorganism.

### 3.5. Fungal Development

When it comes to fungi, NOXs/RBOHs participate in a wide range of biological processes from their growth, differentiation and reproduction, to rhizobial colonization. In *Aspergillus nidulans,* NoxA plays crucial roles in fungal physiology and differentiation by generating ROS [60]. Moreover, genetic analysis of *Δnox2* (lacking the NADPH oxidase 2 gene), *Δnox1* (lacking the NADPH oxidase 1 gene) and a transcription factor deletion mutant *Δste12* in *Sordaria macrospora*, reveals that the mutation of NOXs could lead to ascospore germination defect [61]. In yeast, NoxA, NoxB and their associated regulators (BemA, Cdc24 and NoxR) have distinct or overlapping functions for the regulation of different hyphal morphogenesis processes [57]. Furthermore, in *Neurospora crassa,* NOX-1 elimination results in complete female sterility, decreased asexual development and reduced hyphal growth; whereas, a lack of NOX-2 does not affect any of these processes but led to the production of sexual spores that failed to germinate [62]. In this study, the function of NOX-generated ROS acting as critical cell differentiation signals highlights a novel role for ROS in the regulation of fungal growth [62]. The function of NOX-generated ROS in regulating the reproductive process was also found in other fungi. In *Botrytis cinerea*, NOX complexes are essential for conidial anastomosis tubes’ formation and fusion [63]. 

### 3.6. Other Aspects 

Besides functioning in the regulation of pollen germination, root development and seed germination, as well as fungal development, NOXs/RBOHs have other effects as well. It was suggested that solar ultraviolet irradiation regulates anthocyanin synthesis in apple peel by modulating the production of ROS via NADPH oxidase [64]. Moreover, chloroplastic NADPH oxidase-like activity mediates perpetual H_2_O_2_ generation, which probably induces apoptotic-like cell death of *B. napus* leaf protoplasts [65]. Furthermore, it was reported that among the seven homologues of NADPH oxidases in potato, the expression of *StRbohA* and *StRbohB* was detected in particular when dormancy break [66]. These results suggest very extensive roles of NOXs/RBOHs in the plant kingdom, participating in various important biological processes.

## 4. NOXs/RBOHs in Stress Tolerance

### 4.1. Roles in Plant Immunity

In the traditional sense, biotic stress response works as plant immunity. Besides the external barriers of the plant cell wall, the innate immune system of plants plays an indispensable role in plant immunity under biotic stresses. There are two layers of plant immune recognition. The first layer of innate immunity is initiated by the perception of pathogen-associated molecular patterns (PAMPs) by the surface-localized pattern recognition receptors (PRRs), leading to PTI (PAMP-triggered immunity) [67]. As many pathogens have acquired the ability to inject virulence effector proteins into host cells to achieve more effective infection, the second layer of plant immune recognition that involves intracellular immune receptors forms. The intracellular immune receptors that are most often the nucleotide-binding domain and NLR (leucine-rich repeat containing receptor) proteins can recognize those effectors and elicit a second layer of defense, defined as ETI (effector-triggered immunity) [68]. Correlational studies are mostly carried out in *Arabidopsis*, tobacco and rice. 

It was reported that *Arabidopsis AtRbohD* and *AtRbohF* gene expressions are differentially modulated by PAMPs, which contributes to fine-tune NOX/RBOH-dependent ROS production and signaling in *Arabidopsis* immunity [69]. Moreover, AtRbohD exists in complex with the receptor kinases ELONGATION FACTOR-TU RECEPTOR (EFR) and FLAGELLIN SENSING 2 (FLS2), which are the PRRs for bacterial EF-Tu and flagellin, respectively. The plasma-membrane-associated kinase Botrytis-induced kinase1 (BIK1), which is a direct substrate of the PRR complex, directly interacts with and phosphorylates RbohD upon PAMP perception [70]. It was also reported that AtRbohD triggers death in cells damaged by fungal infection but simultaneously inhibits death in neighboring cells through the suppression of free SA (salicylic acid) and ET (ethylene) levels [71]. In addition, AtRbohF appears to be a key player not only in HR (hypersensitive type of resistance) -related cell death but also in regulating metabolomic responses and resistance [72,73]. RbohD-dependent ROS accumulation stimulates autophagosome formation and limits HR-related cell death [72,73]. A double mutant of *atrbohf* and *atrbohd* resulted in an almost complete loss of resistance to AG8 (a pathogenic bacterium), while single NADPH oxidase mutant had no such effect, suggesting that AtRbohs may correlate and function together in the resistance to some pathogens [74]. Furthermore, overexpressed StRbohA, a potato NADPH oxidase, in *Arabidopsis* enhances the ROS-mediated defense mechanisms [75]. Transient and stable expression of a DN (a dominant negative form) of the *Arabidopsis* AtRop1 in potato reduces pathogen development, which is associated with increased NOX/RBOH-mediated H_2_O_2_ production and JA (jasmonate) signaling [76]. 

In tobacco, *Nicotiana benthamiana* L. NbRbohB is responsible for both the transient PTI ROS burst and the robust ETI ROS burst. It is worth mentioning that MAPKs are involved in the ETI ROS burst by transactivation of NbRbohB, but not in the PTI ROS burst [77]. However, the potato NtRbohD-mediated H_2_O_2_ production does not seem determinant for either hypersensitive response development or the establishment of acquired resistance but it is most likely involved in the signaling pathways associated with the protection of the plant cells [78,79]. In rice, OsRbohA and OsRbohE were reported as participating in ROS-dependent immune responses [80]. In addition, inoculation of rice with *Xanthomonas oryzae* pv. oryzae (Xoo) strain PXO99 can improve the expression levels of two rice NOX genes, *OsRbohA* and *OsRbohB*, further suggesting that the *NOX**/RBOH* genes take part in plant immunity [68]. Moreover, rice blast disease initiated by pathogen invasion needs NADPH oxidase to produce ·O_2_^-^ [81]. In barley, it was reported that HvRbohF2 plays an important role in penetration resistance to *Blumeria graminis* f. sp. *hordei* and the control of leaf cell death [82], while HvRbohA facilitates cellular accessibility to the fungus [83]. All these results further suggest the importance of NOXs/RBOHs in plant immunity.

NOXs/RBOHs also function in elicitor-induced and pathogen-induced stomatal closure [84,85]. The bacterial pathogen *Pseudomonas syringae* pv. tomato synthesizes the polyketide toxin coronatine, which inhibits stomatal closure by MAMPs (microbe-associated molecular patterns) and by ABA. In this process, inhibition of NOX/RBOH-dependent ROS synthesis in guard cells plays an important role [86]. Several studies further indicate that the addition of ATP can trigger ROS production in an RbohD-dependent manner and lead to the rapid closure of leaf stomata in response to bacterial pathogen infections [85,87,88,89,90]. In addition, NOX-dependent ROS synthesis by pathogen helps the pathogen infect plants [81,91] and a coordinated balance of generating, sensing and detoxification of ROS serves a key role in pathogen colonization and fungal virulence [86,92,93]. NOXs/RBOHs also positively regulate *Rhizobium* infection and its release into the nodule cells [94,95]. These results suggest that NOXs/RBOHs play vital roles in both response to the invasion of pathogen and facilitation of *Rhizobium* infection in plants.

### 4.2. Roles in Abiotic Stress Tolerance

NOXs/RBOHs also widely participate in the responses of plants to a number of abiotic stresses, which include salt, hypoxia, heavy metals, drought, wounding and extreme temperature stresses, and play a fundamental role in the stress tolerance.

#### 4.2.1. Salinity Stress

Salinity stress, which induces both ionic and osmotic damages, seriously impairs plant growth, ions uptake and yield production [96]. Plants are equipped with a series of defense responses against salinity stress, such as regulation of ion transport including Na^+^ and K^+^, accumulation of compatible solutes and expression of stress-related genes [96,97]. Maintaining cellular Na^+^/K_+_ homeostasis is pivotal for plant survival in saline environments. By comparing the *Arabidopsis* double mutants *atrbohD1/F1* and *atrbohD2/F2* with wild-type and the single null mutant *atrbohD1* and *atrbohF1* plants in response to NaCl treatments, Ma et al. [98] proposed that ROS produced by both AtRbohD and AtRbohF function as signal molecules to regulate Na^+^/K_+_ homeostasis, thus improving the salt tolerance of the plant. In addition, the double mutants were shown to be more sensitive to salt and less efficient for K_+_ selective uptake [99]. The early H_2_O_2_ generation by the NOXs under salt stress could be the beginning of a reaction cascade that triggers the antioxidant response in plants to overcome the subsequent ROS production, thereby mitigating the salt stress-derived injuries. Further, it was identified that expression of *AtRbohD* and *AtRbohF* are highly induced under salinity stress, and AtRbohF plays a role in the regulation of xylem loading of Na^+^ to protect leaves from salinity stress in *Arabidopsis* [96]. In addition, it was suggested that salinity-induced elevation of [Ca^2+^]_cyt_ corresponds to the plasma membrane (PM) Ca^2+^ influx, as well as Ca^2+^ release from intracellular Ca^2+^ stores plays important roles in ROS signaling and salt tolerance. Moreover, salinity-induced Na^+^ accumulation in the cytosol triggers [Ca^2+^]_cyt_ elevation, leading to activation of NOXs/RBOHs and apoplastic H_2_O_2_ accumulation [96]. AtRbohs contributing to H_2_O_2_ production in response to NaCl or mannitol stress can also increase proline accumulation in plants [100]. In addition, mild salt stress (10 mM NaCl) stimulates biphasic increases in *AtRbohD* expression and ROS production, whereas, in the increased tolerance to *NaCl1 (itn1)* mutant, *AtRbohD* expression is suppressed, which is accompanied by a corresponding reduction in ROS accumulation under salt stress [101,102], suggesting that ITN1 may negatively regulate AtRbohD-dependent ROS production to balance the function of ROS. Moreover, AtRbohD-dependent ROS production transmits long-distance signals in salt stress [103] and may play a role in regulating rapid systemic acquired acclimation (SAA) or systemic response caused by abiotic stresses [102,104]. 

There are also some studies carried out in other plant species. In a living cell of spinach (*Spinacia oleracea* L.) leaf, a large quantity of ROS could be produced by NOXs/RBOHs in epidermic tissue under bisulfite stress [105]. Moreover, it was suggested that pumpkin-grafted cucumber plants increase their salt tolerance via a mechanism involving the root-sourced NOX/RBOH-dependent H_2_O_2_ production, which enhances Na^+^ exclusion from the root and promotes an early stomatal closure [106]. NaCl-induced ROS production can also be decreased by the introduction of a NADPH oxidase in *Torulopsis glabrata*, resulting in a significant increase in growth under hyperosmotic stress conditions [107]. Similarly, ROS-generating activity of NOXs/RBOHs is negatively affected by NaCl in maize [108]. All the results address the significant role of NOXs/RBOHs in salt stress defense of plants.

#### 4.2.2. Hypoxic Stress 

Submergence has an adverse effect on internal oxygen availability, sugar status and survival. Plants suffer from oxygen deficiency (hypoxia) and energy starvation under flooding conditions. Higher plants have evolved complex adaptive mechanisms to flooding that are induced by changes in the cellular redox state and phytohormones [109]. It was reported that the transcript levels of five *Arabidopsis RBOH* genes (*AtRbohA*, *B*, *D*, *G* and *I*) are increased under hypoxia and the transcript levels of *AtRbohD* are significantly increased, in particular at an early stage in the hypoxia response; meanwhile, ethylene modulates H_2_O_2_ signaling via regulation of the expression of the NOX/RBOH genes in the hypoxia pathway [110]. A more recent study further explained the role of NOXs/RBOHs in hypoxia response [111]. The expression of *AtRbohI* can also be increased by hypoxia, and AtRbohI could regulate the expression of genes involved in ethylene biosynthesis and downstream of hypoxia signaling [109]. In rice, the differential transcriptional expression of *NOX/RBOH* genes was also observed under submergence. The regulation of *OsRbohs* expression and ROS production maintaining a homeostasis are helpful to rice seedlings facing different levels of submergence [112]. For example, *OsRbohH* expression under oxygen-deficient conditions is greater in cortical cells than in cells of other root tissues. In addition, the ROS production by OsRbohH, which is induced by CDPK5 and/or CDPK13, is essential for ethylene-induced aerenchyma formation in rice roots under oxygen-deficient conditions [113].

#### 4.2.3. Heavy Metal Stress

Cadmium (Cd), lead (Pb), nickel (Ni) and arsenic (As) are the main environmental pollutants that can cause a great increase in ROS production of plants. Under Cd stress, several NOXs/RBOHs differentially regulate ROS metabolism, redox homeostasis and nutrient balance in *Arabidopsis thaliana* [114]. Similar results were reported in other studies. Seedlings of pea (*Pisum sativum* L.) treated with Cd(NO_3_)_2_ show a growth reduction in all organs, and the activities of NADPH oxidase are increased in root protein extracts [115]. In sunflower leaves, Cd treatment can regulate NADPH oxidase activity and the putative NOX/RBOH gene expression [116]. Moreover, NADPH oxidase responding to Cd stress might be involved by affecting the activity of PM H^+^-ATPase (a protein localized in the PM that can produce a proton-motive force) [117]. A further study indicates that H^+^-ATPase and NADPH oxidase are key factors in activation of some elements of the brassinosteroid (BR)-induced pathway under Cd stress [118]. Pb and Ni have also been reported to have the ability to stimulate oxidative burst in plants. In *Vicia faba* L. roots, Pb treatment could trigger a rapid and dose-dependent increase in ROS production by NOX-like enzyme [119]. Ni induces ROS production in wheat roots and Ca^2+^ may be involved in the NADPH oxidase-mediated ROS production under Ni stress [120]. In addition, NOXs are also involved in the response of plants to As toxicity. Arsenic can induce the uptake and translocation of P, S, Cu, Zn and Fe in the wild-type plants, while the *AtRbohC*-deficient plants show quite lower uptake levels of the nutrient elements such as Zn, Fe, Cu, P and S but higher uptake levels of K, suggesting that NADPH oxidase could be critical in regulating the transport and translocation of As and macro/micronutrients [121].

#### 4.2.4. Wound Response

A little early study in tomato (*Solanum lycopersicum* L.) shows that the NOX/RBOH-supplied ROS intermediates the wound response of the plant [122]. After that, a potato NADPH oxidase gene, *StRbohA*, was found to be responsible for the wound-induced oxidative burst in potato tubers. Loss of its expression increases the susceptibility to microbial infection and contributes to the age-induced loss of wound-healing ability [123]. In *Arabidopsis*, AtRbohD is required for efficient local and systemic wound-induced ROS production [103]. Moreover, it was indicated that the transcripts of *NaRbohD* in *Nicotiana attenuata* are rapidly and transiently elicited by wounding, and are amplified when *Manduca sexta* oral secretions are added to the wounds, suggesting that NaRbohD-based defenses play an important role in late herbivore-elicited responses of a plant [124].

#### 4.2.5. Temperature Response

Many studies have reported that NOXs/RBOHs could respond to cold stress or high temperature. In maize, the activity of NADPH oxidase was found to be increased after 2 h of cooling of seedlings but reversed to the control level after 24 h of cooling [125]. In *Arabidopsis*, low-temperature treatment can induce expression of the *AtSRC2* gene in roots, which is in proportion to the levels of ROS produced partially by AtRbohF. AtSRC2 is an activator of Ca^2+^-dependent AtRbohF-mediated ROS production and may play a role in cold responses [126]. Similarly, a recent study proposed that the expression levels of *FvRbohA* and *FvRbohD* in strawberry are quickly induced by cold, followed by an increase in NADPH oxidase activity [127]. Conversely, at high temperature (30 ℃), the total NADPH oxidase activity and *NtRbohD* expression are all suppressed in tobacco mosaic virus (TMV)-inoculated leaves of tobacco, which resulted in less ROS production and enhanced susceptibility to pathogen and suppression of HR-type necrosis [128,129].

#### 4.2.6. Drought Stress

It was indicated that NADPH oxidase is involved in the ABA-induced production of ROS in maize. When the maize seedlings were subjected to water stress, ABA was accumulated and then triggered the NOX/RBOH-mediated ROS production, resulting in the induction of antioxidant defense systems against oxidative damage in the plants [130]. In rice, drought can induce the total activity of NADPH oxidases [131]. Further, it was found that the transcript of *OsRbohA* was stimulated by drought, ablation of *OsRbohA* impairs the tolerance of plants to various water stresses, whereas its overexpression enhances the tolerance [21]. However, more recently, a study showed that the transcription level of a NADPH oxidase gene, *FRbohD*, and content of H_2_O_2_ are increased in a drought-sensitive cultivar of fescue (*Festuca arundinacea* Schreb) under drought, but lower levels of *FRbohD* transcription and H_2_O_2_ concentration in leaves might contribute to drought stress tolerance for a drought-tolerant cultivar [132]. These results suggest critical roles of NOXs/RBOHs in plant drought tolerance and enhanced expression of the protein genes by bioengineering approaches may improve the stress tolerance.

#### 4.2.7. Hormonal Response

As a kind of stress-responsive protein, the functional characteristics of NADPH oxidase in plants under a number of phytohormonal treatments have been widely investigated. For example, NADPH oxidase activity of plants could be stimulated by the active auxins 2,4-dichlorophenoxyacetic acid (2,4-D), indole-3-acetic acid (IAA) and α-naphthaleneacetic acid (α-NAA) [133], while the activity is unaffected by benzoic acid and the inactive auxin analogs 2,3-dichlorophenoxyacetic acid (2,3-D) and β-naphthaleneacetic acid (β-NAA) [134], suggesting the plant NOXs are auxin-responsive. According to our recent study, the *Arabidopsis NOX/RBOH* genes, *AtRbohB* and *AtRbohD*, are strongly upregulated by methyl jasmonate (MeJA) treatments but markedly downregulated by ABA, zeatin and IAA treatments; whereas, almost all the rice *NOX* genes exhibit great upregulation under hormonal treatments such as ABA, MeJA and salicylic acid (SA) [7]. In pear, the transcription of most PbRbohs could also be regulated by ABA, MeJA and SA treatments [135]. Then, the relationships between ABA level, ROS generation as well as *NOX/RBOH* gene expression in senescing leaves of rice are further clarified, in which *OsNox2*, *OsNox5*, *OsNox6* and *OsNox7* are probably involved in the ABA-induced ·O_2_^-^ generation in the initial stage of leaf senescence. Conversely, *OsNox1*, *OsNox3* and *OsFRO7* are not associated with ABA-induced ·O_2_^-^ generation during leaf senescence [136]. In another study, ABA was suggested to induce the rapid accumulation of ROS via NOXs/RBOHs in suspension culture cells of tobacco, and NOXs/RBOHs and H_2_O_2_ appear to be important components in the ABA signal transduction pathway in plants [137]. Moreover, treatment with ABA significantly increases the activity of NADPH oxidase and a crosstalk between Ca^2+^ and ROS originated from NOXs leads to the induction of antioxidant enzyme activity in maize [138]. NOXs/RBOHs were also reported to have the responsibility to react to ethephon. According to the study, ethephon-mediated leaf senescence, H_2_O_2_ elevation and senescence-associated gene expression are all alleviated by NADPH oxidase inhibitor DPI and reduced glutathione, suggesting that ROS generated from NOXs/RBOHs likely serves as an oxidative stress signal participating in ethephon signaling [139]. The rapid and pronounced responses of NOXs/RBOHs to hormonal treatments, conferred their wide but powerful roles for multi-stress tolerance of plants.

## 5. Regulations of NOXs/RBOHs in Plants

### 5.1. Transcriptional Regulation

It is well known that transcriptional regulation is a fundamental mechanism in all cellular systems. It is mediated by interaction between transcription factors (TFs) and specific *cis*-acting regulatory elements (CAREs) of the promoter sequence from target genes. We have systemically analyzed the *cis*-elements in the promoters from the members of the whole *NOX/RBOH* gene family in *Arabidopsis*, rice and wheat, and elaborately constructed their tissue and inducible expression profiles [7,8]. Analysis for the *cis*-regulatory elements shows that there are more than 30 stress or hormone responsive elements in the *NOX/RBOH* gene promoters randomly distributing in the promoter sequences, suggesting different expression regulatory patterns of the *NOX/RBOH* genes. The unique tissue- or development-specific expression patterns, diverse inducible expression profiles and distinct promoter activities [7,8,19] further indicate the complex spatio-temporal transcriptional regulations between the *NOX/RBOH* family gene members under different circumstances. Another study also identified many *cis*-elements in *Arabidopsis* and rice *NOX/RBOH* gene promoters, which have different response behaviors to various life regimes such as oxidative stress, increased calcium, hormones application as well as cellular differentiation and growth [140]. In addition, a range of *cis*-elements responsible for different kinds of ROS have been dissected in *Arabidopsis* using microarray expression data which can be grouped into two categories: common ROS-related and ROS-specific *cis*-elements [141,142]. Common ROS-related elements involve GCN4_motif, TATCCAT/C_motif and G-Box, while W-box is one of the main ROS-specific elements. GCN4_motif may play a role in reproductive growth and development, while TATCCAT/C_motif may function under starvation. G-Box and other motifs such as ABRE (the ABA-responsive element), TGA-element (the auxin-responsive element), ERE and GCC-box (the ethylene-responsive elements), GARE-motif, P-box and TATC-motif (the GA-responsive elements), CGTCA-motif and TGACG-motif (the JA-responsive elements) as well as TCA element (the SA-responsive element) are also widely distributed in *NOX/RBOH* family genes which are involved in hormone signaling, responses to light and other biological processes, including biotic and abiotic stresses, leaf senescence, and seed development [140]. W-box, as one of the main ROS-specific elements, mainly functions in pathogen responses by binding with WRKY (a protein that binds preferentially to W box (TTGACC/T) on the sequence of promoters) transcriptional factors. It has been reported that MAPKs are instrumental for transcriptional reprogramming by directly or indirectly controlling the activity of transcription factors following PAMP perception [143,144,145]. For example, WRKY8 is phosphorylated by MAPKs, and then binds to the W-box in the *NbRbohB* gene promoter to positively regulate the expression of *NbRbohB* in *Nicotiana benthamiana*. The promoter analysis of *StRbohC*, a tobacco *NbRbohB* ortholog, shows that W-box *cis*-element regulates the gene expression [146]. Consistently, Yoshioka et al. [77] reported that W-box in *NbRbohB* promoter is the target sequence of WRKY transcription factors. Further research showed that the transcriptional activity of WRKY8 is dependent on SIPK and WIPK (SIPK, salicylic acid–induced protein kinase; WIPK, wound-induced protein kinase) by phosphorylation. The phosphorylated WRKY8 increases its DNA transcriptional activation and binds to the W-box sequence [147]. Together, these results imply that MAPKs-mediated phosphorylation of WRKYs perhaps is a general matter endowing the transcriptional activation of WRKYs to bind to and regulate the expression of *NOX/RBOH* genes, and the conserved W-box *cis*-element in the sequence of their promoters serves as a scaffold linking WRKYs.

### 5.2. Activity Regulation

A large number of studies have shown that the functions of NOXs/RBOHs in plants are regulated by lots of signaling particles including Ca^2+^, receptor-like protein kinases (RLKs), receptor-like cytoplasmic kinases (RLCKs), calcium-dependent protein kinases (CDPKs) [70,148,149], BIK1 [70], mitogen-activated protein kinase (MAPK) cascades [149,150,151,152], open stomata 1 (OST1) [153], POP/RAC small GTPases [154,155] and hormones (like ABA and ET). It is obvious that the activity regulation of NOXs/RBOHs is multiple and complex, being in the center of many important signal transduction pathways of plants. 

#### 5.2.1. Ca^2+^-Regulated Protein Kinases-Mediated Regulation 

Ca^2+^ has long been recognized as an essential and conserved secondary messenger in plants, contributing to signaling transduction. According to recent studies, Ca^2+^ exerts important functions in the process of NOX/RBOH-mediated signaling in plants. Plant NOXs/RBOHs usually have two or more EF-hand motifs in the hydrophilic N-terminal regions, suggesting that their activation is dependent on Ca^2+^. In rice, a structural and biochemical analysis not only verified the fact but further figured out that Ca^2+^ only binds directly to the first but not the second EF-hand motif that OsRbohB possesses [156]. Therefore, a certain concentration of Ca^2+^ in the cytoplasm is a prerequisite for the activation of NOXs/RBOHs. In addition, Ca^2+^ binding can also activate CDPKs to phosphorylate OsRbohB for ROS production [157]. In turn, ROS generated at the plasma membrane are presumably released into the apoplastic spaces, where the resultant ROS (H_2_O_2_) then activate the hyperpolarization-activated Ca^2+^ channels to facilitate Ca^2+^ influx causing the second Ca^2+^ elevation in the cytosol [41,157,158,159]. Therefore, there is a circulatory pathway of positive regulation of NOXs/RBOHs for ROS generation during polar cell growth or hypoxia response. The OsRbohB-dependent ROS production is necessary for Ca^2+^ influx, and then the induced ROS in turn may trigger Ca^2+^ efflux from intracellular Ca^2+^ stores in vivo. Similar results were also obtained in other studies [160,161]. These results showed that H_2_O_2_-mediated plasma membrane Ca^2+^ influx perhaps is a general mechanism for maintaining intracellular Ca^2+^ dynamic balance and laying a certain molecular foundation for the activation of NOXs/RBOHs.

Studies performed on other plant species also demonstrate that Ca^2+^ and its regulated protein kinases could activate NOXs/RBOHs and regulate ROS accumulation in plants. For instance, NADPH oxidase from tomato and tobacco can be activated by adding Ca^2+^ in vitro, suggesting that elicitor-induced ROS production by plant NOXs/RBOHs might be dependent on Ca^2+^ [162]. RbohC/RHD2 is required for Ca^2+^ influx via the stimulation of Ca^2+^ channels and for the generation of a tip-focused gradient of cytosolic free calcium [Ca^2+^]_cyt_ in root hairs of *Arabidopsis*, which is essential for polar growth [41]. In addition, another study shows that the conformational change conducted by Ca^2+^ binding to the EF-hand region and direct phosphorylation of AtrbohD synergistically activate its ROS producing activity [158]. As the primary Ca^2+^-regulated kinases, CDPKs become the focus for dissecting the activity and phosphorylation of NOXs/RBOHs. Increasing evidence shows that AtCDPK4, AtCDPK5, AtCDPK6 and AtCDPK11 are the positive regulators of AtRbohD under flg22 treatment [148]. Among these CDPKs, AtCDPK5 was found to be involved in the phosphorylation of Ser39, Ser148, Ser163 and Ser347 of AtRbohD [79]. Similar studies were also carried out in tomato, tobacco and potato, suggesting that phosphorylation regulation of NOXs/RBOHs in a CDPK-dependent pattern widely exists in various species [156,163,164]. Intriguingly, CDPKs perhaps also play negative roles in the regulation of NOXs/RBOHs activity. For example, OsCPK12 (also, namely, OsCDPK9) promotes the tolerance of rice to salt stress by reducing the accumulation of ROS. At the same time, overexpression of OsCPK12 repressed the expression level of OsRbohI [165,166]. Therefore, Ca^2+^ and CDPKs may regulate the activity of NOX/RBOH respectively or synergistically.

In addition to CDPK/CPK, there is another Ca^2+^-regulated kinase represented by calcineurin B-like (CBL)-interacting protein kinases (CIPKs), which becomes activated upon interaction with CBL Ca^2+^ sensor proteins [167,168]. Recent studies demonstrate that tomato CIPK6 interacts with CBL10 and its in vitro activity is enhanced in the presence of CBL10 and Ca^2+^. CBL10 and CIPK6 contribute to ROS generation by direct phosphorylation of NbRbohB during ETI [169]. In *Arabidopsis*, CIPK26 specifically interacts with the plasma membrane-localized Ca^2+^ sensors, CBL1 and CBL9, and they work together to phosphorylate the N-terminus of AtRbohF for ROS production [170]. Moreover, many studies have shown that CBL-CIPK complexes contribute to the tolerance of plants to various abiotic stresses such as salt, cold and drought [171,172,173]. However, although the activation mechanisms for AtRbohD and AtRbohF are similar, AtRbohD has significantly greater ROS-producing activity than AtRhohF [174], implying their functional diversity, at least in part. A further study shows that protein phosphorylation is a prerequisite for Ca^2+^-dependent activation of the NOXs/RBOHs, and so NOX/RBOH phosphorylation is the initial trigger for the plant Ca^2+^-ROS signaling network [174]. These results identify an interconnection between CBL-CIPK-mediated Ca^2+^ and ROS signaling in plants, and Ca^2+^-binding and phosphorylation mediated by CDPK/CIPK synergistically activate the ROS-producing enzyme activity of NOXs/RBOHs in the Ca^2+^-regulated signaling pathway. 

#### 5.2.2. RLKs and RLCKs-Mediated Regulation

Receptor-like kinases (RLKs) belong to a very large protein family with crucial roles in plant growth control, development regulation, cellular signal recognition and transduction. The typical RLKs include three major domains: an extracellular domain, a transmembrane domain, and an intracellular kinase domain; receptor-like cytoplasmic kinases (RLCKs) only possess a cytoplasmic kinase domain [175,176]. Interestingly, it becomes increasingly clear that RLKs are key PRRs for recognition of PAMPs, and the RLCKs often associate with RLKs functionally and physically to relay intracellular signaling via trans-phosphorylation events [177]. Many studies have shown that some RLKs and/or RLCKs can interact with NOXs/RBOHs directly or indirectly and phosphorylate the proteins for transmitting pathogen signals during plant immunity [178,179]. The major signaling pathways are also been summarized in Figure 2, Figure 3 and Figure 4. 

It was reported that botrytis-induced kinase1 (BIK1), a protein of the RLCKVII subfamily member, can directly bind to multiple RLKs/PRRs in the resting state, such as flagellin sensing 2 (FLS2), elongation factor-Tu receptor (EFR) and PEP (Phosphoenolpyruvic acid) 1 receptor (PEPR1/2), which recognize a conserved 22 amino acid peptide of bacterial flagellin (flg22), a conserved N-terminal peptide sequence of the bacterial elongation factor-Tu (termed elf18 or elf26), and the endogenous AtPep1 (an endocrine peptide), respectively [180,181,182]. These RLKs all associate with the regulatory LRR-receptor kinase, BRI1-associated receptor kinase 1 (BAK1) (also known as SERK3). AtRbohD-dependent ROS production in response to flg22 or elf18 is abolished in *fls2* or *efr* mutants [183], implying that FLS2/EFR is required for the ROS production from AtRbohD during plant response to flg22/elf18. Two other RLKs, PEPR1 and PEPR2 (belong to the membrane-localized receptor kinases), coupling with the shared receptor LRR-RK BAK1 in a pH-dependent manner, can act as receptors of AtPeps, a kind of endogenous peptide. AtPeps can elicite immune responses similar to those induced by pathogens [184]. *Arabidopsis* chitin-elicitor receptor kinase1 (CERK1), which is independent of BAK1 or related molecules [185], directly binds chitin through its LysM domain by homodimerization [186]. Actually, OsCERK1 not only participates in the activation of NOXs/RBOHs during chitin-induced immunity [185,187] but also in the activation during bacterial peptidoglycan (PGN)-induced immunity. In addition, other RLK members, like malectin-like receptor kinase FERONIA (FER) and ANX1/2 (ANXURs), are also involved in plant immunity, while functioning in other biological processes such as development and reproduction [188]. FER promotes the association of FLS2 and BAK1 in response to flg22 and elf18 treatments, while ANX1 and possibly ANX2 negatively regulate PTI by putatively competing with FLS2 for interaction with BAK1 [189,190]. That is to say, FER and ANX1 regulate FLS2-BAK1 complex formation in opposite ways. Together, FLS2, EFR, PEPR1/2, FER, ANXs and CERK1 are the upstream PRRs participating in the regulation of NOX/RBOH-mediated ROS production in plant immune response. In *Arabidopsis*, BAK1 serves as a co-receptor of FLS2 for flg22 recognition. BAK1 and FLS2 locate on plasma membrane in FLS2-BAK1 heterodimer form, which act as ‘molecular glue’ involved in immune signaling [191]. FER also acts as a RALF (rapid alkalinization factors)-regulated scaffold that modulates receptor kinase complex assembly [189]. RALFs, rapid alkalinization factors, are the secreted peptides. The members, RALF23/33, can repress FER signaling during PTI responses [189], whereas LLG1 as a cooperator of FER facilitates the formation of FER-FLS2-BAK1 ligand-induced receptor complex for immunity [190]. The FER-FLS2-BAK1 receptor complex can activate BIK1 directly by BAK1, and subsequently, BIK1 phosphorylates RbohD for ROS production to trigger immune responses. This signaling pathway might also trigger immune responses via MAPKs and WRKY46 [190].

In addition, increasing members of RLCKs, such as PBS1-like kinase (PBL1), brassinolide-signaling kinase 1 (BSK1), pattern triggered immunity compromised receptor-like cytoplasmic kinase (PCRK1) and botrytis induced kinase 1 (BIK1), were also identified as mediating the BAK1-dependent PTI signaling in *Arabidopsis* [180,181,182,192]. Among the receptor kinase members, both BAK1 and BIK1 become the focus of multiple signals (Figure 2). For example, AtRbohD-dependent ROS production is compromised in *bak1* or *bik1* mutants in response to both flg22 and elf18 treatments [182,193]. When flg22 or elf18 were recognized by FLS2 or EFR respectively, the latter rapidly recruited the co-receptor BAK1 (or other SERKs: SOMATIC EMBRYOGENESIS RECEPTOR KINASES), and then induced trans-phosphorylation events between BIK1 and the FLS2-BAK1 complex [193,194]. To be clearer, BIK1 is a substrate of BAK1, whereas BAK1 and FLS2 are also substrates of BIK1, for phosphorylation [180]. After that, the rapidly phosphorylated BIK1 (GTP-BIK1) directly phosphorylates AtRbohD to produce ROS [182,195]. PBL1, a highly homologous receptor-like cytoplasmic kinase of BIK1, as the molecular chaperone of BIK1, is also important for PTI signaling, potentially via interactions with FLS2 at rest state [187]. BIK1 and PBL1 play a positive role in the RbohD-dependent ROS production but are not required for MAPK activation [181,196]. Besides controlling RbohD, BIK1 and PBL1 are also required for the PAMP- and/or DAMP-triggered cytosolic Ca^2+^ burst prior to ROS production [178,197,198]. By contrast, the exact roles of the other RLCKs in plant signaling are not as clear as BIK1 and PBL1. For example, BSK1 and PCRK1 are also genetically required for PAMP-triggered ROS burst and may thus also directly phosphorylate RbohD, but more systematic biochemical and functional characterization is required to fully address their role in PTI signaling.

As discussed above, the regulation imposed by BIK1 is a common regulatory mechanism for RbohD, in which BIK1 is the central immune regulator for multiple signaling from upstream to downstream to trigger an oxidative burst from NOXs/RBOHs (Figure 2). Therefore, given the importance of BIK1 in immune signaling, its protein stability is tightly regulated to ensure appropriate and robust immune activation. A previous study reported that BIK1 stability is positively regulated by heterotrimeric G proteins and negatively regulated by the CPK28 [198,199]. A more sophisticated study reveals that PUB25 and PUB26 (two E3 ligases homologous) are the crucial factors which directly target BIK1 for degradation by the ubiquitin proteasome system. The activity of PUB25/26 is negatively or positively regulated by the directly binding with G protein or the phosphorylation from CDPK28, respectively. Interestingly, PUB25/26 specifically target non-activated BIK1 (GDP-BIK1), suggesting that activated BIK1 (GTP-BIK1) is maintained for immune signaling [195]. In addition, a member of MAP4K, SIK1, was recently identified to be able to phosphorylate and enhance BIK1 stability at a resting state by coupling with heterotrimeric G proteins [200]. 

The immune response is also mediated by CERK1 in conjunction with two OsCEBiP homologs, LYM1 and LYM3, which can directly bind to PGN [208], but do not contribute to chitin signaling in *Arabidopsis* [201]. However, a recent study demonstrated that LysM-containing receptor-like kinase5 (LYK5) binds chitin at a much higher affinity than CERK1 [202]. LYK5 is genetically required for a chitin-induced response and forms a chitin-dependent complex with CERK1 [202,209]. Coincidently, PBL27 (the *Arabidopsis* homolog of BIK1) was found to connect the chitin receptor complex CERK1-LYK5 and a MAPK cascade. PBL27 can phosphorylate and activate AtMAPKKK5 [203,204]. These findings provide a phospho-signaling pathway: AtCERK1-PBL27-AtMAPKKK5-AtMKK4/5-AtMPK3/6, in plant immunity. Whether LYK5 and CERK1 are organized into a sandwich-type receptor system similar to CEBiP and CERK1 in rice (Figure 3), and whether the receptor complex LYM1-LYM3-CERK1 activating downstream signaling components was also mediated by AtPBL27, remains to be clarified. As shown in Figure 2, all these receptors and coreceptors are finally associated with NOXs/RBOHs activation directly or indirectly, implying the highly complicated regulatory mechanisms of NOXs/RBOHs in plant immunity.

Two RLCKs in rice, namely OsRLCK185 and OsRLCK176, the homologs of BIK1, can be directly phosphorylated by OsCERK1 during PGN- and/or chitin-induced immunity, and they may also participate in the activation of NOXs/RBOHs and triggering MAPK cascades [201,210,211]. Moreover, the phosphorylation of OsRLCK176 and OsRLCK185 by OsCERK1 is associated with OsLYP4 or OsLYP6 under PGN induction, but associated with OsLYP4, OsLYP6 or CEBiP (LysM-RLP, CHITIN ELICITOR-BINDING PROTEIN) under chitin treatment, suggesting OsLYP4 and OsLYP6 as dual functional PRRs sensing bacterial PGN and fungal chitin (Figure 3) [212]. Meanwhile, different patterns of hetero-oligomeric receptor complexes are assembled under the induction of different ligands. In other words, although LYP4 associates with LYP6, as well as with CEBiP, these complexes partially dissociate following ligand detection [211]. In addition, CEBiP can form a homodimer upon chitin binding that is followed by heterodimerization with CERK1, creating a signaling-active sandwich-type receptor system [213,214]. As shown in Figure 3, the phosphorylated OsRLCK185 by OsCERK1 not only participates in downstream of the activation of NOXs/RBOHs, but also enhances chitin-induced activation of OsMPK3 and OsMPK6 [187]. Further studies more clearly show that the phosphorylation of OsRLCK185 by OsCERK1 can trigger a MAPK signaling cascade: OsMAPKKK18/24-OsMKK4-OsMPK3/6 [215,216,217]. Similar to OsRLCK185, the phosphorylated OsRLCK176 by OsCERK1 might also regulate multiple defense responses triggered by PGN and chitin, such as the activation of MAPKs, ROS generation from NOXs/RBOHs and the expression of defense-related genes [211]. Therefore, the phosphorylated OsRLCK185 and OsRLCK176 can trigger downstream immune responses by activating both MAPK cascades and NOXs/RBOHs-mediated ROS production, but the functional mechanism for how the two RLCKs activate NOXs for ROS production, remains unclear.

Recruitment of regulatory receptor kinases seems to be specified by the type of PRR ectodomain. Accordingly, BAK1 is dispensable for chitin-triggered responses, whereas CERK1 does not participate in flg22-mediated signaling [185,218]. However, CERK1 also associates with BIK1 and is required for chitin-induced responses [181]. As such, a connection is established between chitin- and flg22-mediated signaling. In addition, the activated BIK1 phosphorylates the residues Ser39, Ser339, Ser343 and Ser347 within the N-terminal part of RbohD, while both BIK1 and CPKs phosphorylate Ser347 [67], implying that there is a crosstalk between BIK1 and CPKs-mediated regulation of RbohD activity.

#### 5.2.3. Rho-Type GTPases-Mediated Regulation 

Plants have a subfamily of Rho GTPases, which belongs to the Rat sarcoma (Ras) superfamily of small GTP-binding proteins and is called ROPs (Rho of plants) or RACs (for the sequence similarity shared with animal Racs, a Rho subfamily). The modules of ROP/RAC-NOX/RBOH mediate multiple extracellular signals ranging from hormones, pathogen elicitors and abiotic stresses, and regulate diverse cellular processes which are important for polarized cell growth, differentiation, development and reproduction (Figure 3 and Figure 4) [154,223]. For example, the activated OsRac1 interacts with the N-terminus of OsRbohB and thereby stimulates ROS generation [222]. Duan et al. [223] also proposed that ROP activity is correlated with NOX/RBOH-catalyzed ROS accumulation during polar root hair and pollen tube growth. Furthermore, structural analyses of OsRbohB, coupled with in vitro binding and NMR (nuclear magnetic resonance) titration assays, showed that OsRac1 binds the coiled-coil of the N-terminal region of OsRbohB [67,156,223]. In *Arabidopsis*, it was reported that AtROP1, an OsRac1 homolog, plays an important role in AtRbohH/AtRbohJ-regulated pollen tube growth [32], while AtROP2 is required for AtRbohC/RHD2-dependent ROS formation during root hair growth [41]. 

The ROP/RAC small GTPases serve as a two-state molecular switch depending on its GDP- or GTP-bound conformation [224]. Recently, emerging evidence demonstrates that guanine nucleotide exchange factor (GEF) enhances the release of GDP from ROP/RAC, thereby promoting the binding of GTP. Moreover, GEFs exclusively exert their actions in large molecular complexes linking RLKs to the activation of small GTPase [28,223,225]. FER (FERONIA) as an upstream regulator for the ROP/RAC-signaled pathway that controls RbohC-dependent ROS-mediated root hair development. Moreover, FER can be pulled down by ROP2 GTPase in a guanine nucleotide-regulated manner, implying a dynamic signaling complex involving FER, a ROPGEF and a ROP/RAC [223]. Further, it was reported that the glycosylphosphatidylinositol-anchored protein (GPI-AP) LORELEI (LRE) and the seedling-expressed LLG1 physically interact with FER. They are not only co-transported with FER from the endoplasmic reticulum to the plasma membrane, but also exist as components of the FER-ROPGEF-ROP/RAC-NOX/RBOH signaling pathway [226]. Here, the rapid alkalinization factors (RALFs), play the positive roles of activating (RALF1 during root elongation but RALF23 during seedling growth) [227] (Figure 4) but not repressing (RALF23/33 during PTI responses but RALF22/23 during salt tolerance) FER signaling (Figure 2) [189,228]. 

Once activated, the GTP-bound ROP/RACs could recruit NOXs/RBOHs to mediate downstream ROS-dependent processes of plant growth and reproduction [226]. In addition, ANX1 and 2, the closest homologues of FER, may activate ROP GTPases through RopGEFs, preceding the activation of the NOX/RBOH-catalyzed ROS accumulation during pollen tube growth [28]. In brief, with the cooperators LLG1 (LRE) and RALFs, FER/ANX-GEF, being the upstream regulators, activate ROP/RACs directly, and the activated ROP/RACs subsequently activate NOXs/RBOHs for ROS production to regulate plant growth and reproduction. To sum up, all the results, combined with those mentioned earlier, demonstrate that the membrane receptor complex FER(ANX)-FLS2-BAK1 is not only involved in BIK1-mediated activity regulation of NOXs/RBOHs during plant immune response, but also participates in ROP GTPases-mediated activity regulation of NOXs/RBOHs during normal plant development and reproduction.

In rice, the most typical elicitor in RLK/GEF/RAC-mediated regulation of NOXs/RBOHs signal pathways during plant immunity is chitin. Chitin is one of the best characterized PAMPs in pathogenic and non-pathogenic fungi. Instead of FER in *Arabidopsis*, two PRRs, OsCEBiP and OsCERK1, in which OsCEBiP acts as a receptor-like protein (RLP), directly binds to chitin. The two immune proteins could form a receptor complex to transduce the chitin signals to the downstream components upon fungal infection [201,213]. Here, a homolog of *Arabidopsis* GEF, OsRacGEF1, which was identified as the specific GEF for OsRac1, mediates the change between the Rac-GDP inactive form and the Rac-GTP active form as a molecular switch [204]. OsRacGEF1 interacts with the cytoplasmic domain of OsCERK1, and is directly phosphorylated by OsCERK1 [179]. Besides OsRacGEF1, there is another regulator factor, OsRACK1 (a rice homolog of human Receptor for Activated C-kinase 1), which interacts with OsRac1 [219]. OsRACK1 functions as a scaffolding protein linking heterotrimeric G proteins and MAPK cascade [220], and therefore, OsRac1 plays a role at the downstream of heterotrimeric G protein [229]. The OsRac1 activated by chitin or heterotrimeric G protein then directly regulates OsNOXs/RBOHs for ROS production. These results imply a model OsRacGEF1-OsRac1-mediated signaling involved in the activation of NOXs/RBOHs in rice (Figure 3).

As mentioned above, phosphorylation plays important roles in activating NOXs/RBOHs and oxidative burst. However, the phosphorylation is not sufficient to fully activate NOXs/RBOHs [221]. The conformational change in the EF-hand region of NOXs/RBOHs induced by Ca^2+^ binding to the EF-hand motifs might be essential. It was reported that Ca^2+^ binding to the EF-hand motifs and phosphorylation by CDPK synergistically activate AtRbohD to produce ROS [158]. This prompts us to speculate that a crosstalk between multiple signaling components might play crucial roles in full activation of NOXs/RBOHs in plants. In rice, it was suggested that at the initial stage of the oxidative burst, CDPKs, in a Ca^2+^-dependent manner, may sensitize NOXs/RBOHs for activation by phosphorylating their N-terminal region, thus inducing a conformational change and exposing the site for interaction with Rac GTPase [222]. Moreover, in cytosol, Ca^2+^ flux has diphase change and plays a dual role in NOX/RBOH-mediated oxidative burst. In the initial stage of oxidative burst, cytosolic Ca^2+^ influx may be required by CDPK to phosphorylate NOXs/RBOHs and recruit Rac GTPases, both of which synergistically contribute to ROS burst. In turn, ROS and other components induce more Ca^2+^ influx, and the elevated Ca^2+^ in cytosol induces its efflux from the intracellular stores. When Ca^2+^ accumulation reaches a threshold, it may serve as a negative feedback mechanism to terminate the OsRac1–OsRbohB interaction and ROS production from OsRbohB [222]. These results imply that CDPKs and RACs synergistically regulate the activity of NOXs/RBOHs in a Ca^2+^-dependent manner. Furthermore, the Ca^2+^-related negative feedback mechanism exerts an important function in regulating the RAC–NOX/RBOH interaction and maintaining the homeostasis of ROS in plants. Interestingly, a negative regulatory model was reported in vertebrates. In which, HACE1, a HECT domain and Ankyrin repeat domain containing E3 ubiquitin-protein ligase 1, directly targets Rac1 for degradation of the preformed signaling complex, and therefore negatively regulates Rac1-dependent NADPH oxidases in a ubiquitin-dependent manner [230]. Whether a similar negative regulatory model exists in plants for the RACs-mediated regulation of NOXs/RBOHs, needs further studies. 

#### 5.2.4. MAPK Cascades-Mediated Regulation 

MAPK cascades are one of the most important and highly conserved signaling cascades, which consist of three tier components, MAPKKKs, MAPKKs and MAPKs, widely participating in plant growth and development, as well as in responses to a number of biotic and abiotic stresses [243,244,245,246]. It is well known that both abiotic and biotic stress factors can activate MEKK1 (a MAPKKK), which subsequently initiates a signaling of MEKK1-MKK1/2/4/5-MPK3/4/6 in *Arabidopsis* and acts as upstream of NOXs/RBOHs to stimulate H_2_O_2_ production [151,247,248,249]. In addition, the activation of MPK3 and MPK6 by flg22 is not affected in the *rbohD* mutant, suggesting that these MAPKs act as the upstream of RbohD-mediated oxidative burst in *Arabidopsis* [250,251]. Similar results are also found in *Nicotiana benthamiana* [150] and maize [152]. Coincidentally, SIK1, a mitogen-activated protein kinase kinase kinase kinase (MAP4K) family member, was found to directly interact with and phosphorylate RbohD to promote the extracellular ROS burst upon flagellin perception. Moreover, SIK1, similar to BIK1, associates with the PRR FLS2 at a resting state, and interacts with and stabilizes BIK1 by direct phosphorylation at rest state [200]. In other words, SIK1 positively regulates immunity not only by binding to and activating RbohD directly, but also in an indirect BIK1-mediated way. In yeast and humans, MAP4Ks can directly activate MAPK cascades [205,206]. In addition, in parallel with ROS production, PTI also induces MAPK activation [191,207]. These results prompt us to speculate that two pathways mediated by MAPK: elicitor-MAP4K (SIK1)-RbohD and elicitor-MAP4K (SIK1)-MAPK cascades–NOX/RBOH, also execute important roles in NOXs/RBOHs regulation.

#### 5.2.5. Hormone-Mediated Regulation

There is now a substantial body of studies concerning hormones that participate in plant immunity and abiotic stress responses, such as ABA, JA, SA and ET. [148,237,252,253]. Intrinsic to their participation in plant stress responses is the interplay between ROS and these hormones, as well as hormone-dependent ROS balance through the regulation of NOX/RBOH-based ROS-producing activity and ROS-scavenging capacity. The entire picture addressing this regulation has been summarized in Figure 4. 

ABA has been the focus of intense investigation since it was identified in the 1960s as an endogenous small molecule growth inhibitor and regulator of plant stress physiology. It was reported that NOX/RBOH-dependent ROS production in guard cells plays an important role in ABA-mediated stomatal closure [231], and ABA-induced ROS accumulation mainly originates from two NOX proteins, AtRbohD and AtRbohF, during stomatal closure [232]. Stomatal closure in guard cells is a basic defensive strategy of plants to prevent biotic and abiotic stresses. OST1 is a member of the sucrose non-fermenting 1 (SNF1)-related protein kinase 2 family (SnRK2s). A mutation in the *OST1* gene impairs ABA-triggered ROS production in guard cells, suggesting that OST1 acts upstream of NOX/RBOH in this signaling cascade [233]. In addition, further experiments proved that flg22 treatment could induce stomatal closure in wild-type plants but not in the *ost1* mutant of *Arabidopsis* [232], and OST1-mediated ROS generation in guard cells involves the phosphorylation of AtRbohF by OST1 [153]. In this process, ABA can be perceived by the pyrabactin resistance protein 1 (PYR1), then the PYR1 receptor complex leads to suppression of protein phosphatase 2Cs (PP2Cs), which function as negative regulators of OST1 [235]. The OST1 mediate phosphorylation of AtRbohF, but the protein might also be phosphorylated during the signaling transduction [153]. Considering the highly conserved serine residues on other NOX/RBOH proteins, it is reasonable to believe that they can be phosphorylated by OST1 and/or other members of the SnRK2 family kinase proteins in the regulation of stomatal movement of plants. Up to now, it has been widely accepted that in the absence of ABA, the activated ABI2, a key member of PP2C protein kinases, interacts with and inhibits a SnRK2-type protein kinase by arresting SnRK2 and forming PP2C–SnRK2 (ABI2-SnRK2) complexes, which act as a positive regulator of ABA responses. Increased levels of ABA induce an interaction of PP2C with PYR/PYL/RCAR receptors that triggers the deactivation of PP2Cs to release active SnRK2s [234], thereby activating downstream signaling, such as OST1-mediated phosphorylation of AtRbohF for stomatal closure [153].

Besides OST1 and the associated components, a number of components in ABA signaling have also been characterized, including G proteins, protein kinases/phosphatases and receptors, involved in regulation of NOXs/RBOHs for ROS production. For example, ABA stimulates phospholipase D (PLD) activity probably through a putative ABA receptor. On ABA inhibition of stomatal opening, PLDa1 binds to GPA1 (α-subunit of heterotrimeric G protein) and regulates its function by promoting the conversion of GTP-bound Gα to a GDP-bound Gα, thus producing PA [238]. Subsequently, the second messenger PA directly binds to and activates AtRbohD to regulate ROS production [237]. Moreover, during the ABA-mediated stress response, MAPK cascades may act both upstream and downstream of the ROS production. For instance, a study in maize revealed that ABA activates a 46 kDa MAPK, which not only acts downstream of H_2_O_2_ production, but also positively regulates NOX/RBOH [151]. Another intriguing case shows that FER mutation can cause ABA hypersensitivity in stomatal closing [234], implying that a crosstalk between the FER-mediated growth-promoting and ABA-mediated growth-inhibiting signaling pathways perhaps exists. A further study indicates that the ROP/RAC GTPases ROP11/ARAC10 not only interact with several GEFs such as GEF1, GEF4 and GEF10, but also interact with and activate the phosphatase activity of ABI2 (ABA insensitive 2) after being activated by FER-GEFs [234]. As mentioned above, when the concentration of ABA increases, it induces PYR/PYL/RCAR receptors to interact with and arrest PP2C, and subsequently triggers the deactivation of PP2Cs to release active SnRK2s for stomatal closure [234]. On the other hand, ABI2 can directly interact and dephosphorylate FER, providing a negative feedback mechanism for FER-mediated signaling [236]. Therefore, ABI2 acts as a center regulation factor involved in the regulation of ABA signaling and FER signaling by interacting with PYR/PYL/RCAR receptors and PP2Cs or by interacting with ROP/ARAC and FER, respectively. This may be the crucial point as to why FER-mediated signaling can inhibit the ABA pathway. ABA- and stress-hypersensitive responses in *fer* mutant can be partially rescued by the *abi2-1* gain-of-function mutation.

Increasing evidences also reported that an ethylene receptor 1 (ETR1) and ethylene insensitive 2 (EIN2)-mediated signaling is required for the AtRbohD-dependent ROS accumulation during flagellin-induced intracellular response [148]. AtPep1, a 23-amino acid endogenous peptide, which was initially identified as a DAMP in *Arabidopsis* [240], plays a critical role in flagellin-inducing plant immunity [241]. Both ET and Pep1 treatments can induce BIK1 phosphorylation by Pep1 receptor kinases, PEPR1 and also likely PEPR2, which, when coupled with the shared receptor LRR-RK BAK1 [184], can trigger an innate immunity in *Arabidopsis* [182]. It is to say, similar to AtPeps-triggered responses, ethylene-induced ROS production from activated RbohF depends on BIK1-mediated phosphorylation by Pep1/2 receptor kinases. This prompts us to speculate that ET accumulation in plants under the treatments of exogenous factors including ET and AtPep1, can lead to endogenous AtPeps accumulation, which can be perceived by the receptor PEPR1/1 and then triggers BIK1-mediated NOX/RBOH phosphorylation for ROS production and stomatal closure.

ET not only potentiates the accumulation of ROS, but also together with the produced ROS, regulates stress-induced cell death [254]. RbohF, but not RbohD, mediates ethylene-induced ROS production and stomatal closure. Interestingly, RbohF seems to not be involved in flg22-mediated stomatal closure [148,239]. Moreover, a number of studies have shown that immune responses serve to amplify PTI signaling [182,240,255,256] via the pathways of JA/ET and/or SA [240,256]. JA seems to enhance all AtPep-triggered responses [184]. In addition, treatment with MeJA contributes to H_2_O_2_ production through AtrbohD and AtrbohF in a coronatine-insensitive1 (COI1) depended way [242]. COI1 is a LRR (Leu-rich repeat)/F-box motif-containing protein that determines the substrate specificity of the SCF-type E3 ubiquitin ligase [257]. In addition, another study shows that BR can also enhance NADPH oxidase activity and thereby elevates H_2_O_2_ levels in apoplast [258]. Although these results highlight the roles of plant hormones in the regulation of NOXs/RBOHs activity during plant development and resistance, the mechanisms and associated components still remain to be addressed further.

#### 5.2.6. Other Kinds of Particles-Mediated Regulation

Besides those common signaling particles mentioned above, there are also other various kinds of particles functioning in the regulation of NOXs/RBOHs activity. For instance, ATP-mediated stomata close is NOX-dependent. Addition of ATP leads to the rapid closure of leaf stomata and enhances resistance of plants to the bacterial pathogen, *Psuedomonas syringae*. At the same time, a L-type lectin receptor-like kinase (LecRKI.9) DORN1, the receptor of ATP, can directly phosphorylate NADPH oxidase RbohD, while mutation of DORN1 phosphorylation sites on RbohD eliminates the ability of ATP to induce stomatal closure [84]. In addition, a 14-3-3 protein, identified as an interactor, plays a positive role in inducing ROS production from NtRbohD [67,259]. Moreover, it was found earlier that the activity of NOXs/RBOHs can be inhibited by cAMP [260]. It was reported that the NADPH-oxidase activity is finely regulated spatially and temporally by cellular signaling events that trigger the translocation of the cytosolic subunits to its membrane partner involving post-translational modifications and activation by second messengers, such as arachidonic acid (AA), while trans-AA isomers inhibit NADPH-oxidase activity by direct interaction with enzyme components [261]. Moreover, a report shows that one of the biological effectors, nitric oxide (NO), regulates the activities of NOX proteins by S-nitrosylation [262]. In vitro analysis further showed that AtRbohD can be specifically *S*-nitrosylated at Cys890 of the C-terminal portion by NO and *S*-nitrosylation could inhibit AtRbohD-dependent ROS production in response to *Pst* DC3000 expressing effector AvrB [263]. Therefore, it is not hard to notice that, with the development of the studies on the regulation of NOXs/RBOHs, more and more particles will be identified to be associated with their activity, even though the complete relationship of those particles in the process of NOX/RBOH regulation still remains to be figured out.

## 6. Conclusions and Perspectives

Currently, NADPH oxidases, being ROS producers, multiple signaling centers and diverse function implementers in plants, are becoming the focus of studies. Increasing members of them have been identified and reported. The highly conservative structure and complex evolution history endow them with stable functions donating ROS and versatile functions, involved in pollen gemination and root development, seed germination, fungal development, symbiosis between plants and microorganisms, stress response, etc. Over the past decade, several important regulation pathways addressing NOX/RBOH activity, such as Ca^2+^-, CDPK-, BIK1-, Rac1- and/or SIK1-mediated signaling, became more and more clear. The high flexibility in NOXs/RBOHs regulation allows plant cells not only to prevent unintended signaling activation but also to modulate signaling amplitude and fine-tune immune responses, as well as to keep development integrity. However, numerous questions still remain to be answered in the future, for instance, the spatiotemporal dynamics of RAC-dependent activation, diphase Ca^2+^ flux change, the crosstalks between different signal pathways, the mechanisms of the messenger ROS transmit signals and NOX/RBOH-based ROS homeostasis and its exact meaning. In addition, increasing yield and improving quality are the foundations of agricultural development and the ultimate aim of scientific research. However, almost all of the results currently obtained are just on the basis of model plants, and only a few field crops have been considered for deep research. Therefore, new breakthroughs in NOX/RBOH research based on field crops such as wheat have become an urgent and charming expectation for sustainable agriculture. 

## Figures and Tables

**Figure 1 cells-09-00437-f001:**
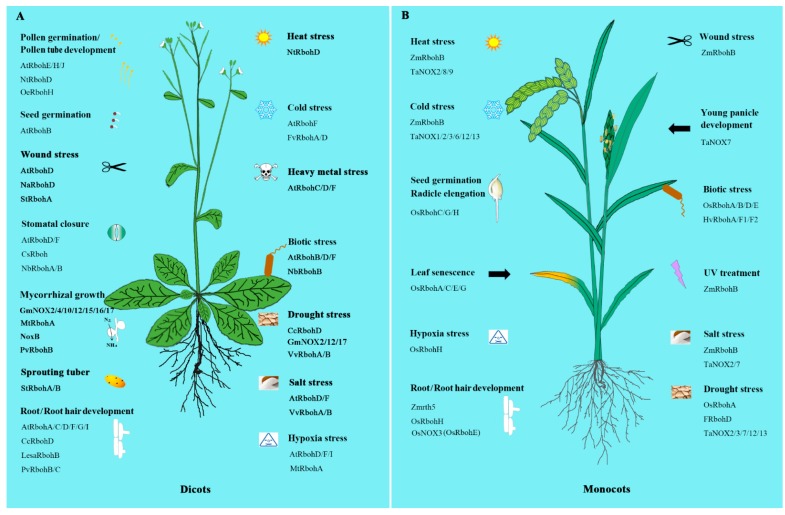
The involvement of NADPH oxidases/respiratory burst oxidase homologs (NOXs/RBOHs) in dicots and monocots. A. The NOXs/RBOHs in dicots. B. The NOXs/RBOHs in monocots. All the acronyms of Rbohs, NOXs, NoxB and rth5 represent the NOX/RBOH homologues from various species. Just as, AtRbohs represent the NOX/RBOH homologues from *Arabidopsis thaliana*, CsRbohs from *Cucumis sativus*, FRbohs from *Festuca arundinacea*, FvRbohs from *Fragaria x ananassa*, GmRbohs from *Glycine* max, HvRbohs from *Hordeum vulgare*, LesaRbohs from *Lepidium sativum*, MtRbohs from *Medicago truncatula*, NaRbohs from *Nicotiana attenuate*, NbRbohs from *Nicotiana benthamiana*, NoxB from *Fusarium oxysporum*, NtRbohs from *Nicotiana tabacum*, OeRbohs from *Olea europaea* L, OsRbohs from *Oryza sativa* L, PvRbohs from *Phaseolus vulgaris*, StRbohs from *Solanum tuberosum*, TaNOXs from *Triticum aestivum*, VvRbohs from *Vitis vinifera*, ZmRbohs and Zmrth5 from *Zea mays*. Please refer to Appendix A for the detailed information of NOXs/RBOHs in the supplementary data.

**Figure 2 cells-09-00437-f002:**
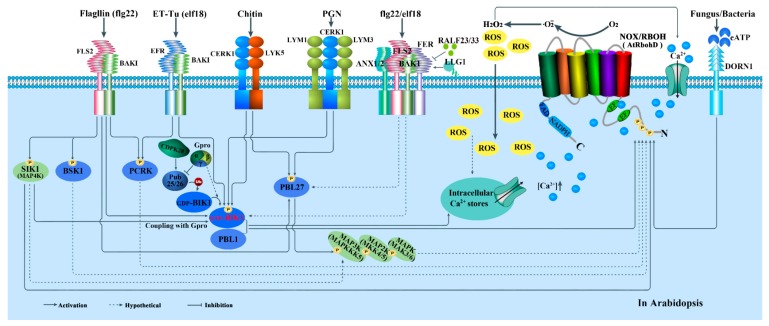
Multiple regulatory signaling pathways of NADPH oxidases (NOXs/RBOHs) in plant immunity of *Arabidopsis*. Botrytis-induced kinase1 (BIK1), a protein of the RLCKVII subfamily member, is the central immune regulator for multiple signaling pathways from upstream to downstream to trigger an oxidative burst from RbohD. BIK1 can directly bind to multiple RLKs/PRRs in the resting state, such as flagellin sensing 2 (FLS2), elongation factor-Tu receptor (EFR), and PEP 1 receptor (PEPR1) [180,181,182]. These RLKs all associate with the regulatory LRR-receptor kinase BRI1-associated receptor kinase 1 (BAK1) (also known as SERK3) and form the immune receptor complexes upon bacterial flagellin (flg22), bacterial elongation factor-Tu (elf18 or elf26) or the endogenous AtPep1 (and related peptides) by interacting with them. When flg22 or elf18 are recognized by FLS2 or EFR respectively, the latter rapidly recruit the co-receptor BAK1 for trans-phosphorylation events between BIK1 and BAK1 [193,194]. Other RLK members, FER (FERONIA) and ANX1/2 (ANXURs), belonging to malectin-like receptor kinases, also known as *Catharanthus roseus* L. receptor-like kinase 1-like proteins (CrRLK1Ls), also participate in NOXs/RBOHs regulation during plant immune response [188]. FER acts as a Rapid Alkalinization Factors (RALF)-regulated scaffold that modulates receptor kinase complex assembly [189], while ANX1 and possibly ANX2 negatively regulate pathogen-associated molecular pattern (PAMP)-triggered immunity (PTI) by putatively competing with FLS2 for interaction with BAK1 [190]. Where, RALF23/33, the secreted peptides, play the negative role in repressing FER signaling during PTI responses [189]. In addition, a LORELEI-like GPI-anchored protein 1 (LLG1), as a cooperator of FER, facilitates FER-FLS2-BAK1 ligand-induced receptor complex formation to activate BIK1, and subsequently phosphorylates the ROS-producing RbohD [190]. Another important immune complex composes of CERK1 (the homologs of BAK1) and two LysM proteins, LYM1 and LYM3 (the homologs of LYP4 and LYP6 in rice), upon recognizing PGN (peptidoglycan) by directly binding to LYM1 and LYM3 [200], but this immune complex does not contribute to chitin signaling in *Arabidopsis* [201]. Meanwhile, a new LysM-containing receptor like kinase 5 was found that binds chitin at a much higher affinity than CERK and forms a chitin-dependent complex with CERK1 [202]. These immune complexes in which AtCERK1 is involved also activate BIK1 by phosphorylation directly. The activated BIK1 (GTP-BIK1) by immune complexes then directly phosphorylates and activates AtRbohD [182,195]. However, AtCERK1-associated PBL27 (an ortholog of OsRLCK185) is not involved in chitin-induced ROS production [203], but it regulates the activation of the MAPK cascade by phosphorylation of AtMAPKKK5 [204] and supervenes with tier phosphorylation events: between AtMAPKKK5 and AtMKK4/5 and AtMPK3/6 in PAMPs signaling [203]. Besides BIK1 and PBL27, there are other RLCKs, such as BSK1, PCRK1, and PBL1, that are also genetically required for a PAMP-triggered ROS burst. PBL1, a close homolog of receptor-like cytoplasmic kinases of BIK1, acting as the molecular chaperone of BIK1, is also important for PTI signaling potentially via interactions with FLS2 at rest state [187]. BIK1 and PBL1 play a positive role in the RbohD-dependent ROS production but are not required for MAPK activation [181,196]. Besides controlling RbohD, BIK1 and PBL1 are also required for the PAMP and/or DAMP-triggered cytosolic Ca^2+^ burst that precedes ROS production [99,178,197]. Just recently, SIK1, a mitogen-activated protein kinase kinase kinase kinase (MAP4K) family member, was found to directly interact with and phosphorylate RbohD to promote the extracellular ROS burst upon flagellin perception. Moreover, SIK1 interacts with and stabilizes BIK1 by direct phosphorylation at rest state [200]. In other words, SIK1 positively regulates immunity not only by binding to and activating RbohD directly, but also indirectly through the BIK1 mediated ways. In yeast and humans, MAP4Ks can directly activate MAPK cascades [205,206]. In addition, in parallel with ROS production, PTI also induces MAPK activation [191,207]. Two regulation pathways of NOX/RBOH activity mediated by MAPK might exist in plant immune response: elicitor-MAP4K (SIK1)-RbohD or/and elicitor-MAP4K (SIK1)-MAPK cascades-RBOH. The regulation imposed by BIK1 is a common regulatory mechanism for RbohD during immune response, and in which BIK1 is the central immune regulator for multiple signaling from upstream to downstream to trigger an oxidative burst from RbohD. Therefore, its protein stability is tightly regulated to ensure appropriate and robust immune activation. PUB25 and PUB26 (E3 ligases homologous) are the crucial factors which directly target BIK1 for degradation by the ubiquitin proteasome system, while the activity of PUB25/26 were negatively or positively regulated by the directly binding with G protein or the phosphorylation from CDPK28, respectively. Interestingly, PUB25/26 specifically target non-activated BIK1 (GDP-BIK1), suggesting that activated BIK1 (GTP-BIK1) is maintained for immune signaling [195]. The member of MAP4K, SIK1, can also phosphorylate directly and enhance BIK1 stability at a resting state by coupling with heterotrimeric G proteins [200]. In addition, ATP can also be released into the extracellular matrix and referred to as extracellular ATP (eATP), functioning in signaling. Its mediated stomata close is NOX-dependent under bacterial and fungal infection, and during the process, a L-type lectin receptor-like kinase (LecRKI.9) DORN1 acts as the receptor of eATP to directly phosphorylate downstream of AtRbohD [85]. BAK1, BRI1-associated receptor kinase 1; BIK1, botrytis-induced kinase1; BSK1, brassinolide-signaling kinase1; CDPK, calcium-dependent protein kinase; CERK, chitin-elicitor receptor kinase; DAMP, damage associated molecular pattern; DORN1, a L-type lectin receptor-like kinase; eATP, extracellular ATP; EFR, elongation factor-Tu receptor; elf18, a bacterial elongation factor-Tu; FLS2, flagellin sensing 2; flg22, a bacterial flagellin; G_pro_, GTP-binding protein; LLG1, LORELEI-like GPI-anchored protein 1; LRR, leucine-rich repeat; LYK, LysM-containing receptor-like kinase; MAPK, mitogen-activated protein kinase; PAMP, pathogen-associated molecular pattern; PBL, PBS-like kinase; PCRK1, pattern-triggered immunity compromised receptor-like cytoplasmic kinase 1; Pep, plant elicitor peptide; PEPR1, phosphoenolpyruvic acid 1 receptor; PGN, peptidoglycan; PRR, pattern recognition receptor; PTI, PAMP-triggered immunity; RALF, rapid alkalinization factors; RLCK, receptor-like cytoplasmic kinases; RLK, receptor-like protein kinase; SIK1, MAP4K, salt inducible kinase 1.

**Figure 3 cells-09-00437-f003:**
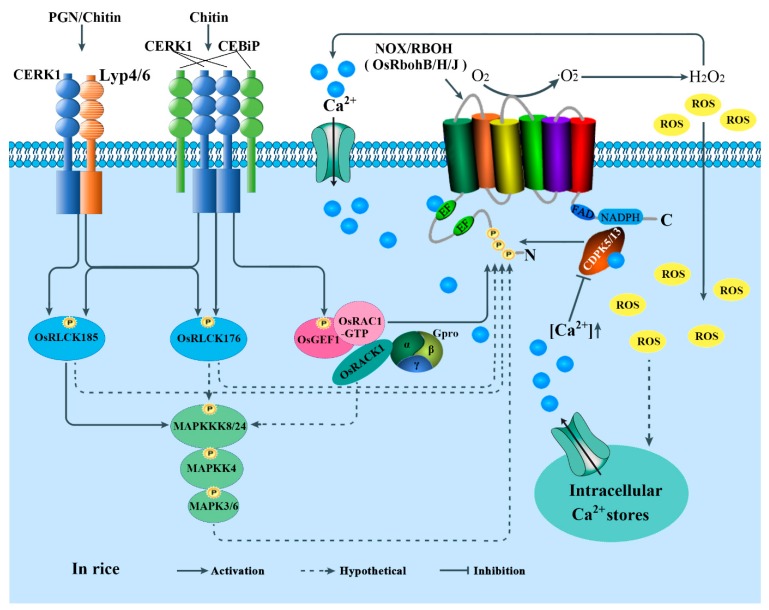
The activity regulation of NADPH oxidases (NOXs/RBOHs) in plant immunity of rice. Two receptor-like cytoplasmic kinases (RLCKs), OsRLCK185 and OsRLCK176, can be directly phosphorylated by a plasma membrane-localized receptor-like kinase, OsCERK1 [201,210,211], which is associated with receptor-like kinases OsLYP4 or OsLYP6 when induced by PGN, but associated with OsLYP4, OsLYP6 or OsCEBiP under chitin treatment [211]. In addition, OsCEBiP can form a homodimer upon chitin binding that is followed by heterodimerization with OsCERK1, creating a signaling-active sandwich-type receptor system [213,214]. Then, the phosphorylated OsRLCK185 triggers a MAPK cascade: OsMAPKKK18/24-OsMKK4-OsMPK3/6, to activate the downstream proteins directly, or perhaps act as the upstream of NOXs/RBOHs to enhance ROS production [215,216]. Similar to OsRLCK185, OsRLCK176 also activates MAPK cascades [211], but the mechanism remains to be addressed as well. A ROP/RAC GTPase, OsRAC1, is also involved in the OsCERK1/OsCEBiP-mediated immune signaling, it activates OsRbohB for ROS production by directly interacting with the N-terminus of the NADPH oxidase [201,213]. Two other regulator factors, OsGEF1 and OsRACK1, also participate in this signaling pathway. OsGEF1 mediates the change between the RAC-GDP inactive form and the RAC-GTP active form as a molecular switch for the signaling [179,204], coupling with OsRACK1 [219], after being phosphorylated by OsCERK1 [28]. In addition, OsRACK1 also functions as a scaffolding protein linking heterotrimeric G proteins and the MAPK cascade [220]. Phosphorylation functions in important roles in activating NOXs/RBOHs and contributes to oxidative burst. However, the phosphorylation of NOXs/RBOHs is not sufficient for full activation of the proteins [221]. The binding of influx Ca^2+^ to EF-hand region and the phosphorylation in the N-terminal region of OsRbohB by CDPK5/13 perhaps induce the conformational change in EF-hand region of the NOX/RBOH and therefore, expose the site for interaction with RAC GTPase in the N-terminal region, and subsequently activate OsRbohB for ROS production. In turn, the ROS produced by the NOX/RBOH and other components induce more Ca^2+^ influx and the elevated Ca^2+^ in cytosolic play a positive role to induce more Ca^2+^ efflux from intracellular Ca^2+^ stores. Meanwhile, Ca^2+^ accumulation reaches a threshold, which may serve as a negative feedback mechanism to terminate the OsRAC1–OsRbohB interaction and ROS production [222]. These results imply that CDPK and RAC synergistically regulate the activity of NOXs/RBOHs in Ca^2+^-dependent manner. Furthermore, the Ca^2+^-related negative feedback mechanism exerts an important function in regulating the RAC–NOX/RBOH interaction for maintaining the ROS homeostasis in rice plants. CDPK, calcium-dependent protein kinase; CEBiP, chitin elicitor-binding protein; CERK, chitin-elicitor receptor kinase; G_pro_, GTP-binding protein; GEF, guanine nucleotide exchange factors; LYP, homologs of LysM proteins in rice; MAPK, mitogen-activated protein kinase; PGN, peptidoglycan; RACK, receptor for activated C-kinase; RLCK, receptor-like cytoplasmic kinases; ROP/RAC GTPase, a subfamily of Rho-type GTPases, which belongs to the Rat sarcoma (Ras) superfamily of small GTP-binding proteins; SIK1/MAP4K, salt inducible kinase 1.

**Figure 4 cells-09-00437-f004:**
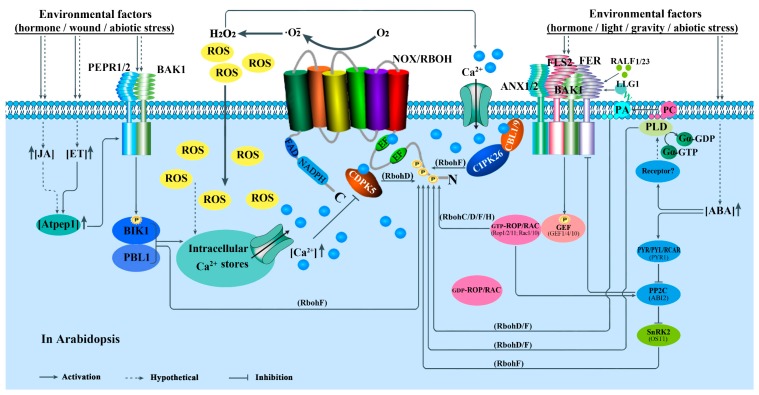
Regulatory signaling pathways of NADPH oxidases (NOXs/RBOHs) in plant development and abiotic stress tolerance. RAC/ROP-NADPH oxidases (NOXs/RBOHs) mediate multiple extracellular signals ranging from hormones, pathogen elicitors and abiotic stresses, and regulate diverse cellular processes important for polarized cell growth, differentiation, development and reproduction [153,223]. Similar to BIKI, ROP/RAC is also recruited by the membrane receptor complex FER (ANX)-FLS2-BAK1. However, the functions of them are different. BIK1-mediated regulation of NOXs/RBOHs is mainly involved in plant immune response, while ROP GTPases-mediated regulation of NOXs/RBOHs mainly participates in plant normal development and reproduction or response to abiotic stresses. LLG1 and RALFs still act as the molecular chaperones of FER to regulate the formation of receptor complex; however, coupling with LLG1, RALFs here play a positive role to facilitate the binding of FER to BAK1 during root elongation (RALF1) or seedling growth (RALF23) [227]. As a molecular switch, GEF is also required for the change between the ROP/RAC-GDP inactive form and ROP/RAC-GTP active form [224]. Once activated, the GTP-RAC/ROPs will interact with NOXs/RBOHs (such as RbohC/D/F/H/J) directly, to mediate downstream ROS-dependent processes of plant growth and reproduction [32,226]. During these processes, the Ca^2+^ binding in EF-hand region and the phosphorylation of NOXs/RBOHs mediated by CDPKs (CDPK26) perhaps can enhance the interaction between GTP-RAC/ROPs and NOXs/RBOHs. In addition to CDPK/CPK, there is another Ca^2+^-regulated kinase represented by calcineurin B-like (CBL)-interacting protein kinases (CIPKs), which becomes activated upon interaction with CBL Ca^2+^ sensor proteins [167,168]. CIPK26 interacts with the plasma membrane-localized Ca^2+^ sensors CBL1 and/or CBL9, they work together to phosphorylate the N-terminus of AtRbohF for ROS production. However, the phosphorylation of NOXs/RBOHs is not sufficient for full activation of the NOXs/RBOHs [221], implying that some additional determinants are required to activate NOXs/RBOHs synergistically with CIPK26. ABA-induced ROS accumulation originates from two NOX/RBOH proteins, AtRbohD and AtRbohF, playing an important role in stomatal closure [231,232]. OST1, a member of the sucrose non-fermenting 1 (SNF1)-related protein kinase 2 family (SnRK2s), is upstream of AtRbohF by phosphorylation [153,233]. Treating with flg22 leads to ABA accumulation in plant cells, which can be perceived by the PYR/PYL/RCAR receptors, such as pyrabactin resistance protein 1 (PYR1), and induce a complex of PYR/PYL/RCAR-PP2C formation. The complex then suppress the activity of A-type protein phosphatase 2Cs (PP2Cs), which releases active SnRK2s [234], such as OST1 [235], and therefore leads to NOXs/RBOHs-mediated ROS production for stomatal closure. In addition, an intriguing case shows that the ROP/RAC GTPases, ROP11/ARAC10, not only interact with several GEFs such as GEF1, GEF4 and GEF10, but also interact with and activate the phosphatase activity of ABI2 (ABA insensitive 2) after being activated by FER–GEFs [234]. On the other hand, ABI2 can directly interact and dephosphorylate FER, providing a negative feedback mechanism for FER-mediated signaling [236]. Thus, a crosstalk mediated by ABI2 links two signaling pathways: the FER involved in growth-promoting and the ABA participating in growth-inhibiting, during regulation of NOXs/RBOHs activity. On ABA inhibition of stomatal opening, PLDa1, a member of phospholipase D, can be activated by binding to GPA1 (a-subunit of heterotrimeric G protein) and regulates its function by promoting the conversion of GTP-bound Gα to a GDP-bound Gα, thus phosphatidylcholine (PC) was hydrolyzed to PA [237,238]. Then, the second messenger PA binds to and activates AtRbohD [237]. Whether the ABA-stimulated PLD activity is mediated through an ABA receptor, needs further verification. In contrast, RbohF-mediated ethylene-induced ROS production and stomatal closure are independent from flg22-mediated stomatal closure [148,239]. In this case, AtPep1, a 23-amino acid endogenous peptide, initially identified as a DAMP in *Arabidopsis* [240], plays a critical role in flagellin-inducing plant immunity [241]. Both ET and AtPep1 treatments can induce BIK1 phosphorylation by Pep1 receptor kinases, PEPR1, and also likely PEPR2. It is to say, similar to AtPep-triggered responses, ethylene-induced ROS production from activated RbohF is dependent on BIK1-mediated phosphorylation by Pep1/2 receptor kinases. Similar to ET, JA and MeJA also seem to enhance the AtPep-triggered responses in plants [184,242]. ABI2, ABA insensitive 2; BAK1, BRI1-associated receptor kinase 1; BIK1, botrytis-induced kinase1; CBL, calcineurin B-like; CDPK, calcium-dependent protein kinase; CIPK, calcineurin B-like-interacting protein kinases; DAMP, damage associated molecular pattern; ET, ethylene; FLS2, flagellin sensing 2; GEF, guanine nucleotide exchange factor; GPA1, G proteinαsubunit 1; JA, jasmonate; LLG1, LORELEI-like GPI-anchored protein 1; MeJA, methyl jasmonate; OST1, open stomata 1; PA, phosphatidic acid; PC, phosphatidylcholine; Pep, plant elicitor peptide; PEPR1, phosphoenolpyruvic acid 1 receptor; PLDa1, phospholipase D alpha; PP2C, protein phosphatase 2C; PYR, pyrabactin resistance protein; PYL, PYR-LIKE; RCAR, regularity components of ABA receptor; RALF, rapid alkalinization factors; ROP/RAC GTPase, a subfamily of Rho-type GTPases, which belongs to the Rat sarcoma (Ras) superfamily of small GTP-binding proteins; SnRK2, sucrose non-fermenting 1 (SNF1)-related protein kinase 2.

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
