# Peer review of "NADPH Oxidases: The Vital Performers and Center Hubs during Plant Growth and Signaling"

_cells, 2020, doi:10.3390/cells9020437_

Round 1

Reviewer 1 Report

The authors have provided an extensive survey of recent publications on NADPH oxidases (NOXs) and related components from a wide range of plants. This will be a valuable resource for researchers and students. Although the review covers a very broad field, the authors have been careful to introduce terms carefully and to define acronyms as far as possible.

Supplementary Material. This is a very helpful catalogue of recent publications. I am surprised that NOX representatives from soybean have not been included - for example “Zhang, Z., et al. (2019) Genomic, molecular evolution, and expression analysis of NOX genes in soybean (Glycine max) Genomics (2019) Volume 111, pp 619-628.” Figure 1, Title. I suggest that this should read. “The involvement of NOXs/RBOHs in dicots and monocots.” In most cases, the exact function of these components is not clear – what are the intracellular and apoplastic targets of ROS, and how does the concentration of ROS influence plant cell development, etc? Figure 1 graphics. It seems to me that this graphic is a rather simplistic generalisation from a limited amount of experimental information. For example, it ignores the fact that, in legumes, the development of the nitrogen-fixing root-nodule symbiosis is influenced by NOX expression. Montiel, J., Fondseca-Garcia, C. and Quinto, C. (2018) Phylogeny and expression of NADPH oxidases during symbiotic nodule formation Agriculture 2018, 8(11), 179; https://doi.org/10.3390/agriculture8110179 Use of English. In general, the manuscript reads very well, but it should be sub-edited to improve the general standard of English. Listed below are a few examples of text that could be slightly misleading.

l. 40 …have been identified…

l.42. ..systemic analysis…

l.43… To date ...

l.53 … signaling systems ...

l.66. ... In fungi, there were four types ...

l.75 … involve the destruction ...

l.97 … kinds, relative to …

l.103 … NOX5-like NOXs ...

l.122… the only protein family identified ...

l.178 uptake of minerals ...

l.231 … in a wide range of …

l.258 … delete “eternal”l. 478 … (ABA), zeatin …

l. 730, 731 … meaning is unclear.

Author Response

Response to the Reviewer 1 Comments

Point 1: Supplementary Material. This is a very helpful catalogue of recent publications. I am surprised that NOX representatives from soybean have not been included – for example “Zhang, Z., et al. (2019) Genomic, molecular evolution, and expression analysis of NOX genes in soybean (Glycine max) Genomics (2019) Volume 111, pp 619-628.”

Response 1: Thanks a lot for your valuable suggestions. We have quoted the important information of GmNOXs from the articles “Zhang, Z., et al. 2019” in the new version and its “Supplementary Material”. Please see Page6, Figure1; Page8, Line 502-503 in the revised text, and in the “Supplementary Material” (SP4-5).

Point 2: Figure 1, Title. I suggest that this should read. “The involvement of NOXs/RBOHs in dicots and monocots.” In most cases, the exact function of these components is not clear – what are the intracellular and apoplastic targets of ROS, and how does the concentration of ROS influence plant cell development, etc? Figure 1 graphics. It seems to me that this graphic is a rather simplistic generalisation from a limited amount of experimental information. For example, it ignores the fact that, in legumes, the development of the nitrogen-fixing root-nodule symbiosis is influenced by NOX expression. Montiel, J., Fondseca-Garcia, C. and Quinto, C. (2018) Phylogeny and expression of NADPH oxidases during symbiotic nodule formation Agriculture 2018, 8(11), 179; https://doi.org/10.3390/agriculture8110179

Response 2: We have use “The involvement of NOXs/RBOHs in dicots and monocots” as the Title of Figure I instead of “The function of NOXs/RBOHs in dicots and monocots”. Please see Page6, Line 374 under Figure1.

Point 3: Use of English. In general, the manuscript reads very well, but it should be sub-edited to improve the general standard of English. Listed below are a few examples of text that could be slightly misleading.

Response 3: We have checked the English throughout the text carefully, refer to the detailed information in the new version. At the same time, some errors you mentioned here have also been corrected.

  1. 40 …have been identified…

Response: We have corrected it in the new version (P. 3, L. 56).

l.42. ..systemic analysis…

Response: We have corrected it in the new version (P. 3, L. 58)

l.43… To date ...

Response: We have corrected it in the new version (P. 3, L. 60).

l.53 … signaling systems ...

Response: We have corrected it in the new version (P. 3, L. 67).

l.66. ... In fungi, there were four types ...

Response: We have corrected it in the new version (P. 4, L. 95).

l.75 … involve the destruction ...

Response: We have corrected it in the new version (P. 4, L. 104).

l.97 … kinds, relative to …

Response: We have corrected it in the new version (P. 4, L. 127).

l.103 … NOX5-like NOXs ...

Response: We have corrected it in the new version (P. 4, L. 133).

l.122… the only protein family identified ...

Response: The sentence “As the protein family identified only in terrestrial plants, typical plant NOXs undergo a complicated evolution” has been revised as follows: “The typical NOX family, being identified only in terrestrial plants, underwent a complex evolution”. The later perhaps will be easier to understand. Please see Page5, Line 194-195 in the new version.

l.178 uptake of minerals ...L280

Response: We have corrected it in the new version (P. 7, L. 411).

l.231 … in a wide range of …

Response: We have corrected it in the new version (P. 8, L. 500).

l.258 … delete “eternal”   

Response: We have corrected it in the new version (P. 9, L. 537).

  1. 478 … (ABA), zeatin …

Response: We have corrected it in the new version (P. 11, L. 831).

  1. 730, 731 … meaning is unclear.

Response: The sentence “Meanwhile, different patterns of ligand-induced hetero-oligomeric receptor complexes are presented when they are of functions.” has been revised as follows: “Meanwhile, different patterns of hetero-oligomeric receptor complexes are assembled under different ligands inducements.” The later perhaps will be easier to understand. Please see P20, L1179-1181 in the new version.

Reviewer 2 Report

The review manuscript entitled „NADPH Oxidases: the Vital Performers and Center Hubs during Plant Growth and Signaling” is a comprehensive piece of cutting edge plant enzymology science and it is obvious the authors have put tremendous labor to collect all the data about NADPH oxidases summarizing it into this manuscript. These data are presented in three main sections where the structure and evolution, function, and regulation of NOXs are elaborated. Moreover, the manuscript has a logical flow which eases readability and keeps reader’s attention. Some, in the first place syntax and grammar corrections, have to be made in order to increase the scientific value of the manuscript:

L8: I believe “henan” should be capitalized.

L18: phrase “A huge of literatures” has strange formulation.

L25: I would put “sophisticated” or maybe more poetic term “elegant” instead of “meticulous”. However, the term “ingenious” has to be replaced, since indirectly points to the creator, which is not scientifically acceptable.

L36: add “plant” before “development”.

L44: add “family” after “NOXs/RBOHs”.

L46-47: phrase “with development stage dependent” is strangely formulated.

L49: advance cannot be in immunity, but in immunology.

L58: last “and” replace with “while”.

L66: put “animals, higher plants”. Instead “There were four type” write “There are four types”.

L75: “in which their functions involve in” – strange formulation.

L78: instead “which comprises” put “which is comprised of a”.

L79: add “a” before “non-“.

L81: you cannot start a sentence with “While”. Start the sentence with “The second group contains four…”

L82-83: “could translocate to the plasma” – vague. What is translocated?

L86: delete “While”.

L87: put “enables” instead of “bridges”.

L88: “Further studies have shown”

L90: write “lasts” instead of “is”.

L93: there is more than one animal kingdom? Start the next sentence with “A mammal genome”

L94-95: after comma, the sentence loses clarity.

L97: “different kinds with Nox1-4:” – please reformulate.

L98: delete “the” before “four”.

L104: “even though”

L105: part of the sentence before comma needs rephrasing.

L110: “elicitor and can”

L121: term “Complex” is more correct here than “Complicated”.

L122: first part of the sentence has wrong syntax. Maybe “underwent” is more convenient than “undergo”.

L123: “plants” instead of “Plantae”.

L125-128: bad syntax. Please rephrase.

L129: “two kinds of algae” – please provide a taxonomical category.

L129-130: the whole sentence – wrong syntax.

L130-131: “For the evolutionary way in plants” – please explain the phrase.

L133: start with “Besides that, domain gain,…”

L135: “perhaps” – have you proved it or not?

L136: “in genetics” – what is that supposed to mean?

L137: delete “the“ before “more”.

L138: I suggest omitting “Functions of” from the subtitle.

L144: Left picture: “hair” instead of “hari”. Right picture: “Young” instead of “Yong”.

L146: I would say you are describing pollen germination or pollen tube development in this section. Hence, the subtitle should be adjusted accordingly.

L149: maybe “pulsating amounts of H2O2”?

L151, 153 & 156: “pollen” not “Pollen”

L157: why did you abbreviate phosphatidic acid? Do you use this abbreviation in further text? If not, such unnecessary abbreviations should be omitted since they hinder reading. There are a lot of examples in the text, so I suggest checking throughout. However, I strongly suggest providing a list of useful abbreviations below the abstract, or where acceptable.

L159: please define “ROP/RAC”.

L167: delete “the” and put “and” between the two genes.

L168: “which implies” instead of “which implying”.   “hrd2” stands for “hair root”, but above is “ROOT HAIR”.   Write “also plays roles in…”

L174: “related” instead of “relating”.

L176: “pollen germination” instead of “pollen development”.

L206: put “Arabidopsis” in italics. And further in the text where isn’t.

L210: “ROS” – different font size.

L214: since other interactions besides symbiosis are described below, I suggest putting “Interactions” instead of “Symbiosis”.

L215: “oxidase” – small caps.

L224: “Fusarium oxysporum” – in italics.

L227: add “and” between “mutualistic” and “symbiotic”.

L228: delete “interactions”.

L230: “fungi” instead of “fungal”. In L243 too.

L231: “plays” instead of “play”.

L241: add “the” before “regulation”.

L246: replace “development” with “germination”.

L249: “chloroplastic” – small caps.

L253: “plant kingdom” is singular. Replace “almost all” with “various”.

L255: like in the L138, I suggest omitting “Functions of”

L258: term “eternal” should be deleted.

L261: add “the” before “surface”

L262, 265, 276, 277 & 284: please reconsider putting abbreviations such as “PTI”, “NLR”, “SA”, “ET”, “HR” and “DN”.

L305: Arabidopsis – italics

L331: a sentence cannot be started with “While”.

L335: “yellow” – why is this important?

L340: do not put comma after “while”.

L343: “pv. tomato” should not been in italics.

L344-345: delete “the hormone”.

L347: “dependent” instead of “depended”.

L349: “helps the pathogen invasion of plants” – sounds like plants are invading someone. Please rephrase.

L352: before “These” put full stop.

L353: “Rhizobium” in italics.

L356: “have been identified function” – please rephrase to make sense.

L357: “heavy metals”.

L361: delete “Therefore”, because plants are not equipped with these mechanisms in order to increase their yield.

L365: delete “(WT)”.

L366. put “[97]” after “Ma et al.” and delete from the sentence end.

L368-369: wrong formulation.

L387: from “other kinds of plant species” delete “kinds of”.

L406: “hypoxia” instead of “hypoxic”.

L416: “arsenic” should not be capitalized.

L419: replace “found” with “reported”.

L420: merge “Cd” with the rest of the formula.

L423: “PM” – what is this standing for?

L427: replace “acts” with “has”.

L430: please put “Arsenic” in the beginning of the sentence instead of “As”.

L431: do not put comma after “while”.

L436: actual scientific name is “Solanum lycopersicum”.

L437: “Late” – what is the meaning of such a sentence beginning?

L442: “attenuata” instead of “attenuate”.

L444: replace “the“ with “a”.

L449: add “the” before “levels”.

L456: “HR-type necrosis” – please define.

L466: you didn’t put species author names anywhere in the text.

L474: do not put comma after “while”.

L474 & 475: delete space between “2,” and “3”.

L478: delete “abscisic acid” and leave just the abbreviation. Misspelled “Zaetin” – it should be “zeatin” (small caps too).

L480: “also could” -> “could also“

L491: “react” instead of “response”.

L492: “DPI” – please define.

L500, 501, and so on: “cis” should stand in italics.

L502 & 509: put “Arabidopsis” in italics also.

L507: delete comma before “further” and replace “complicated” with “complex”.

L510: “different treatments” – I would say “various life regimes” or similar.

L511: maybe to add “increased” before “calcium”, since calcium is not a treatment. “hormone application” too. Also reconsider “cellular development”, since is not a “treatment”.

L515: “is possible the role” – wrong syntax.

L529: Put “[146]” after “Yoshioka et al.” and delete from the sentence end.

L531: “showed” instead of “found”.

L543-544: “and so on” is not an adequate scientific term. You should omit it.

L544: “complex” instead of “complicated”.

L553: full stop before “Therefore”.

L560: replace “acquired” with “obtained”.

L562: “which lays a certain molecular foundation” – please rephrase to get clearer.

L596: remove “the activity of”.

L597: “in vitro”, “in vivo” should stand in italics.

L606: “involved” instead of “involving”.

L610: add “the” before “regulation”.

L612: put semicolon after “ROS”.

L618: delete “kinase”.

L619: “direct phosphorylation of”

L620: “Arabidopsis” in italics.

L634: three times “and” – it should be rewritten.

L636: delete “whereas”.

L656: delete comma before “directly”.

L660: “are also involved” and replace “with also” with “, while”.

L671: remove comma after “whereas”.

L676-678: “brassinolide”, “”pattern” and “botrytis” – small caps.

L680: “become the focus of attention” – please rephrase.

L682: “recruits”.

L684: “clearer”

L691: delete “in”.

L702: replace “who” with “which”.

L703: delete “while”.

L710: remove “in” and “LYM1 and LYM3”.

L714: “homolog”. In parenthesis: “…found to connect the…”.

L717: replace “plant immune” with “plant immunity”. “…CERK1 are organized…”

L724-725: write “…immunity, and they may also…”

L761: put [216] after “Duan et al.” and remove from the sentence end.

L768: “serve as a two-state”

L772: “FER” – please elaborate.

L778: “endoplasmatic reticulum” should not be italicized nor capitalized.

L814: “jasmonate” – small caps.

L815: pleases rephrase the beginning of the sentence.

L825: “normal plant”

L827: “is” instead of “was”.

L828: delete “the”.

L837: please rephrase the part of the sentence starting with “OsRac1”.

L842: “…sufficient to fully activate NOXs…”

L846: “…roles in full activation”

L849: “their N-terminal region,”

L850: “cytosol” instead of “cytosolic”.

L854: write “…in cytosol induce its efflux from the intracellular stores.”

L858: finish the sentence after “manner” and start the next with “Furthermore,”

L870: delete “stress factors” after “abiotic”.

L880-881: relocate “indirect” to stand before “BIK1”.

L887: maybe to replace “literatures” with “reports” or “studies”.

L888: delete “etc”.

L902: “mutant of” instead of “mutant in”.

L913: “complexes which act as”

L922: “α-subunit” instead of “a-subunit”.

L927: add “production” after “H2O2”.

L928: delete “for H2O2 production”.

L938: “involved” instead of “involving”.

L950: delete comma after “that”.

L955: delete “Where,”

L956: add “being” after “not”.

L960: should “pathway” stand at the end of the sentence?

L964: put “highlight” instead of “emphasize”.

L973: replace full stop with comma and continue the sentence.

L976: “earlier”?

L982: strange font size for “I”.

L985: “development of the studies on the”

L900-991: needs to be rephrased.

L993: “complex” instead of “complicated”.

L994: add “germination” after “pollen”.

L1000: “to be answered in the future, for instance...”

L1005: “have been obtained”

L1006: “items exhibiting”

L1007: add “while only” before “few”

Author Response

Response to the Reviewer 2 Comments

Point 1: The review manuscript entitled NADPH Oxidases: the Vital Performers and Center Hubs during Plant Growth and Signaling” is a comprehensive piece of cutting edge plant enzymology science and it is obvious the authors have put tremendous labor to collect all the data about NADPH oxidases summarizing it into this manuscript. These data are presented in three main sections where the structure and evolution, function, and regulation of NOXs are elaborated. Moreover, the manuscript has a logical flow which eases readability and keeps reader’s attention. Some, in the first place syntax and grammar corrections, have to be made in order to increase the scientific value of the manuscript:

Response 1: Firstly, thanks for your pertinent suggestions. We have checked the English throughout the text carefully, please refer to the detailed information in the new version. At the same time, some errors including you mentioned here have also been corrected.

L8: I believe “henan” should be capitalized.

Response: We have changed the “henan” to “Henan” in the new version (P1, L8).

L18: phrase “A huge of literatures” has strange formulation.

Response: We have changed the “A huge of literatures” to “A lot of literatures” in the new version (P1, L18).

L25: I would put “sophisticated” or maybe more poetic term “elegant” instead of “meticulous”. However, the term “ingenious” has to be replaced, since indirectly points to the creator, which is not scientifically acceptable.

Response: We have changed the “meticulous” and “ingenious” to “sophisticate” and “dexterous” respectively, in the new version (P1, L25-26).

L36: add “plant” before “development”.

Response: We have added “plant” before “development” in the new version (P3, L51).

L44: add “family”.

Response: We have added “family” after “NOXs/RBOHs” in the new version (P3, L60).

L46-47: phrase “with development stage dependent” is strangely formulated.

Response: We have changed the phrase “with development stage dependent” to “relying on development stage” in the new version (P3, L63).

L49: advance cannot be in immunity, but in immunology.

Response: We have changed the phrase “abiotic stress responses, and immunity” to “as well as biotic and abiotic stress responses” in the new version (P3, L65).

L58: last “and” replace with “while”.

Response: We have shorten and changed the phrase “deducing that the NADPH oxidases play as the vital performers and center hubs in plant development integrity and signaling, and hoping to” to “hoping to” in the new version (P4, L86-87).

L66: put “animals, higher plants”. Instead “There were four type” write “There are four types”.

Response: We have changed the phrase “animal, higher plant” to “animals, higher plants”; and changed “There were four type” to “In fungi, there are four types” in the new version (P4, L95).

L75: “in which their functions involve in” – strange formulation.

Response: We have changed the phrase “in which their functions involve in” to “in which their functions involve” in the new version (P4, L104).

L78: instead “which comprises” put “which is comprised of a”.

Response: We have changed the phrase “which comprises” to “which is comprised of a” in the new version (P4, L107).

L79: add “a” before “non-“.

Response: We have added “a” before “non-glycosylated” in the new version (P4, L108).

L81: you cannot start a sentence with “While”. Start the sentence with “The second group contains four…” 

Response: We have changed the phrase “While the second comprises four” to “The second group comprises four” in the new version (P4, L110).

L82-83: “could translocate to the plasma” – vague. What is translocated?

Response: We have changed the phrase “The second group could translocate to the plasma membrane to form” to “These four regulatory proteins could be translocate to the plasma membrane and form” in the new version (P4, L111-112).

L86: delete “While”.

Response: We have deleted “While” from the sentence “While the p47phox component……” and changed the sentence to “The p47phox component……” in the new version (P4, L116).

L87: put “enables” instead of “bridges”.

Response: We have changed the phrase “bridges interactions between” to “enables the interactions between” in the new version (P4, L117).

L88: “Further studies have shown”

Response: We have changed phrase “Further studies show” to “Further studies have shown” in the new version (P4, L117).

L90: write “lasts” instead of “is”.

Response: We have changed the word “is” to “lasts” in the new version (P4, L119).

L93: there is more than one animal kingdom? Start the next sentence with “A mammal genome”

Response: We have changed the sentence “Mammal genome ……” to “A mammal genome .…..” in the new version (P4, L123).

L94-95: after comma, the sentence loses clarity.

Response: We have changed the sentence “Mammal genome generally contains seven genes encoding gp91phox homologs: Nox1–Nox5, where gp91phox is renamed Nox2, and the distantly related so called “double oxidases” Duox1 and Duox2” to “A mammal genome generally contains seven genes encoding gp91phox homologs: five close relatives of gp91phox homologs (Nox1–Nox5), and two distant relatives of gp91phox homologs (Duox1 and Duox2)” in the new version (P4, L123-125).

L97: “different kinds with Nox1-4:” – please reformulate.

Response: We have changed the phrase “different kinds with Nox1-4:” to “different kinds relative to Nox1-4:” in the new version (P4, L127).

L98: delete “the” before “four”.

Response: We have corrected it in the new version (P4, L128).

L104: “even though”   

Response: We have corrected it in the new version (P4, L133).

L105: part of the sentence before comma needs rephrasing.

Response: We have changed the sentence “The typical NOXs are conserved structural properties that all possess four conserved domains,……” to “The typical NOXs all possess four conserved domains, ……” in the new version (P5, L176-177).

L110: “elicitor and can”

Response: We have changed “elicitor, can” to “elicitor and can” in the new version (P5, L182).

L121: term “Complex” is more correct here than “Complicated”.

Response: We have put term “Complex” instead of “Complicated” in the new version (P5, L193).

L122: first part of the sentence has wrong syntax. Maybe “underwent” is more convenient than “undergo”.

Response: We have changed the sentence “As the protein family identified only in terrestrial plants, typical plant NOXs undergo a complicated evolution.” to “The typical NOX family, being identified only in terrestrial plants, underwent a complex evolution.” in the new version (P5, L194).

L123: “plants” instead of “Plantae”.

Response: We have put term “plants” instead of “Plantae” in the new version (P5, L195).

L125-128: bad syntax. Please rephrase.

Response: We have changed the sentence “In this model, all FRO family members originated from a common ancestor that contains only Ferric_reduct domain, and the Ferric_reduct domain then obtained FAD_binding_8 and NAD_binding_6 domains by first gene fusion and duplication and clustered into FRO I, FRO II, and FRO III subfamilies.” to “In this model, all FRO family members originated from a common ancestor which contains only Ferric_reduct domain. During evolutionary process, this ancestor obtained FAD_binding_8 and NAD_binding_6 domains by first gene fusion and duplication, and then clustered into FRO I, FRO II, and FRO III subfamilies.”. Please refer to the detailed information in the new version (P5, L197-200).

L129: “two kinds of algae” – please provide a taxonomical category.

Response: We have provided the taxonomical category after algae, such as “two kinds of algae (rhodophytes and chlorophytes)” in the new version (P5, L201-202).

L129-130: the whole sentence – wrong syntax.

Response: We have changed the sentence “FRO I obtained another NADPH_Ox domain to form the typical NOXs in plants.” to “After that, FRO I obtained another important domain-NADPH_Ox and converted into the typical NOXs in plants.” in the new version (P5, L202-203).

L130-131: “For the evolutionary way in plants” – please explain the phrase.

Response: The phrase “For the evolutionary way in plants,” is needless and is removed. Please see the detailed information in the new version (P5, L203).

L133: start with “Besides that, domain gain,…”

Response: "Domain gain" is the way mentioned above, by which FRO I evolve into NOX, so "domain gain" is included into the following several ways of gene evolution. Meanwhile, we refined sentence “Besides domain gain, gene duplication, gene fusion, and exon shuffling might be……” to “In addition to domain gain, gene duplication, gene fusion as well as exon shuffling might be……”. Please refer to the information in the new version (P5, L205).

L135: “perhaps” – have you proved it or not?

Response: Yes, we have proved it, so we removed the word “perhaps” from the text. Please refer to the information in the new version (P5, L208).

L136: “in genetics” – what is that supposed to mean?

Response: We have changed “in genetics” to “in phylogenetics” in the new version (P5, L208).

L137: delete “the“ before “more”.

Response: We have removed “the“ before “more” in the new version (P5, L209).

L138: I suggest omitting “Functions of” from the subtitle.

Response: We have removed “Functions of” from the subtitle in the new version (P5, L210).

L144: Left picture: “hair” instead of “hari”. Right picture: “Young” instead of “Yong”.

Response: We have corrected the two words in the new version (P6, Figure 1).

L146: I would say you are describing pollen germination or pollen tube development in this section. Hence, the subtitle should be adjusted accordingly.

Response: We have changed the subtitle “Pollen Development” to “Pollen germination and pollen tube growth” in the new version (P7, L378).

L149: maybe “pulsating amounts of H2O2”?

Response: We have changed the phrase “pulsating H2O2” to “pulsating amounts of H2O2” in the new version (P7, L381).

L151, 153 & 156: “pollen” not “Pollen”

Response: We have corrected it in the new version (P7, L383, 385, 388).

L157: why did you abbreviate phosphatidic acid? Do you use this abbreviation in further text? If not, such unnecessary abbreviations should be omitted since they hinder reading. There are a lot of examples in the text, so I suggest checking throughout. However, I strongly suggest providing a list of useful abbreviations below the abstract, or where acceptable.

Response: We abbreviate “phosphatidic acid” to “PA” for convenient read in the further text. Thank you for your reminding, we have carefully checked all the abbreviations and removed the some unnecessary abbreviations. Meanwhile, we also provided a list of abbreviations below the abstract. Please refer to the information in the revised text (P2-3).

L159: please define “ROP/RAC”.

Response: ROP/RAC GTPases: Rho-type GTPases belong to the Rat sarcoma (Ras) superfamily of small GTP-binding proteins, and plants have a sole subfamily of Rho-type GTPases, called ROPs (Rho of plants) or RACs (for the sequence similarity they share with animal Racs, a Rho subfamily). Please refer to the information in the revised text (P7, L295).

L167: delete “the” and put “and” between the two genes.

Response: we have deleted “the” and put “and” between the two genes in the revised text (P7, L304).

L168: “which implies” instead of “which implying”.   “hrd2” stands for “hair root”, but above is “ROOT HAIR”.   Write “also plays roles in…”

Response: we have used “which implies” instead of “which implying”, changed “ROOT HAIR” to “hair root”, and written “also plays roles in…”instead of “also play the performers in…”in the revised text (P7, L304-305).

L174: “related” instead of “relating”.

Response: we have written “related” instead of “relating” in the revised text (P7, L311).

L176: “pollen germination” instead of “pollen development”.

Response: we have written “pollen germination” instead of “pollen development”in the revised text (P7, L313).

L206: put “Arabidopsis” in italics. And further in the text where isn’t. 

Response: we have changed all “Arabidopsis” to “Arabidopsis” in the revised text (P8, L357).

L210: “ROS” – different font size.

Response: we have corrected the font size of “ROS” in the revised text (P8, L361).

L214: since other interactions besides symbiosis are described below, I suggest putting “Interactions” instead of “Symbiosis”.

Response: we have written “Interactions” instead of “Symbiosis” in the revised text (P8, L365).

L215: “oxidase” – small caps.

Response: we have changed “Oxidase” to “oxidase” in the revised text (P8, L366).

L224: “Fusarium oxysporum” – in italics.

Response: we have changed “Fusarium oxysporum” to “Fusarium oxysporum” in the revised text (P8, L375).

L227: add “and” between “mutualistic” and “symbiotic”.

Response: we have added “and” between “mutualistic” and “symbiotic” in the revised text (P8, L380).

L228: delete “interactions”.

Response: we have deleted “interactions” in the revised text (P8, L380).

L230: “fungi” instead of “fungal”. In L243 too.

Response: we have deleted “fungi” instead of “fungal” in the revised text (P8, L382; P9, L403).

L231: “plays” instead of “play”.

Response: we have deleted “plays” instead of “play” in the revised text (P8, L384).

L241: add “the” before “regulation”.

Response: we have changed “regulation” to “the regulation” in the revised text (P8, L393).

L246: replace “development” with “germination”.

Response: we have written “germination” instead of “development” in the revised text (P9, L406).

L249: “chloroplastic” – small caps.

Response: we have changed “Chloroplastic” to “chloroplastic” in the revised text (P9, L409).

L253: “plant kingdom” is singular. Replace “almost all” with “various”.

Response: we have changed “plant kingdoms, participating in almost all” to “plant kingdom, participating in various” in the revised text (P9, L409).

L255: like in the L138, I suggest omitting “Functions of”

Response: we have removed “Functions of” from the title in the revised text (P9, L415).

L258: term “eternal” should be deleted.

Response: we have removed “eternal” from the title in the revised text (P9, L418).

L261: add “the” before “surface”

Response: we have changed “surface-localized” to “the surface-localized” in the revised text (P9, L421).

L262, 265, 276, 277 & 284: please reconsider putting abbreviations such as “PTI”, “NLR”, “SA”, “ET”, “HR” and “DN”.

Response: we have written all the abbreviations instead of full spellings of “PTI”, “NLR”, “SA”, “ET”, “HR” and “DN” in the revised text (P9, L421, L425, L435-436, L443).

L305: Arabidopsis – italics

Response: we have changed “Arabidopsis” to “Arabidopsis” in the revised text (P11, L504).

L331: a sentence cannot be started with “While”.

Response: we have changed “While” to “However” in the revised text (P12, L529).

L335: “yellow” – why is this important?

Response: we have changed “yellow pathogenic bacterium strain PXO99” to “Xanthomonas oryzae pv. oryzae (Xoo) strain PXO99” in the revised text (P12, L534).

L340: do not put comma after “while”.

Response: we have changed “while,” to “while” in the revised text (P12, L538).

L343: “pv. tomato” should not been in italics.

Response: we have changed “pv. tomato” to “pv. tomato” in the revised text (P12, L542).

L344-345: delete “the hormone”.

Response: we have changed “by the hormone abscisic acid (ABA)” to “by ABA” in the revised text (P12, L542-543).

L347: “dependent” instead of “depended”.

Response: we have changed “depended manner” to “dependent manner” in the revised text (P12, L546).

L349: “helps the pathogen invasion of plants” – sounds like plants are invading someone. Please rephrase.

Response: we have changed “helps the pathogen invasion of plants” to “helps the pathogen infect plants” in the revised text (P12, L547-548).

L352: before “These” put full stop.

Response: we have changed “, These results” to “. These results” in the revised text (P12, L550).

L353: “Rhizobium” in italics.

Response: we have changed “Rhizobium” to “Rhizobium” in the revised text (P12, L552).

L356: “have been identified function” – please rephrase to make sense.

Response: we have changed the passage “NOXs/RBOHs also widely participate in the responses of plants to a number of abiotic stresses and play a fundamental role in the stress tolerance. Many NOXs have been identified function in the tolerance of plants to salt, hypoxia, heavy metal, drought, wounding, and also extreme temperature stresses.” to “NOXs/RBOHs also widely participate in the responses of plants to a number of abiotic stresses, which including of salt, hypoxia, heavy metals, drought, wounding, and extreme temperature stresses, and play a fundamental role in the stress tolerance.” in the revised text (P12, L554-556).

L357: “heavy metals”.

Response: we have changed “heavy metal” to “heavy metals” in the revised text (P12, L555).

L361: delete “Therefore”, because plants are not equipped with these mechanisms in order to increase their yield.

Response: we have changed “Therefore, plants…” to “Plants…” in the revised text (P12, L559).

L365: delete “(WT)”.

Response: we have deleted “(WT)” in the revised text (P12, L563).

L366. put “[97]” after “Ma et al.” and delete from the sentence end.

Response: we have changed the sentence “Ma et al. proposed that ROS produced by both AtRbohD and AtRbohF function as signal molecules to regulate Na+/K+ homeostasis, thus improving the salt tolerance of the plant [99].” to “Ma et al. [99] proposed that ROS produced by both AtRbohD and AtRbohF function as signal molecules to regulate Na+/K+ homeostasis, thus improving the salt tolerance of the plant.” in the revised text (P12, L559). In addition, it's important to note that we added two serial numbers of references before [97], so [97] becomes to [99].

L368-369: wrong formulation.

Response: we have changed the sentence “In addition, the double mutants show more sensitive to salt and less efficient K+ selective uptake” to “In addition, the double mutants showed more sensitive to salt and less efficient for K+ selective uptake” in the revised text (P12, L566-567).

L387: from “other kinds of plant species” delete “kinds of”.

Response: we have changed “other kinds of plant species” to “other plant species” in the revised text (P13, L604).

L406: “hypoxia” instead of “hypoxic”.

Response: we have changed “by hypoxic” to “by hypoxia” in the revised text (P13, L623).

L416: “arsenic” should not be capitalized.

Response: we have changed “Arsenic” to “arsenic” in the revised text. (P13, L633).

L419: replace “found” with “reported”.

Response: we have written “reported” to “found” in the revised text (P13, L636).

L420: merge “Cd” with the rest of the formula.

Response: we have written “Cd (NO3)2” to “Cd(NO3)2” in the revised text (P13, L637).

L423: “PM” – what is this standing for?

Response: “PM” is the abbreviation of “plasma membrane” in the text (P13, L640).

L427: replace “acts” with “has”.

Response: we have changed the sentence “I In Vicia faba roots, Pb treatment could trigger a rapid and dose-dependent increase in ROS production, and NOX-like enzyme acts a potential role in the ROS production” to “In Vicia faba roots, Pb treatment could trigger a rapid and dose-dependent increase in ROS production by NOX-like enzyme” in the revised text (P13, L643-644).

L430: please put “Arsenic” in the beginning of the sentence instead of “As”.

Response: we have changed “As can induce…..” to “Arsenic can induce….” in the revised text (P14, L658).

L431: do not put comma after “while”.

Response: we have changed “while, the AtRbohC…..” to “while the AtRbohC….” in the revised text (P14, L659).

L436: actual scientific name is “Solanum lycopersicum”.

Response: we have changed “Lycopersicon esculentum” to “Solanum lycopersicum L.” in the revised text (P14, L663).

L437: “Late” – what is the meaning of such a sentence beginning?

Response: we have changed “Late,” to “After that,” in the revised text (P14, L664).

L442: “attenuata” instead of “attenuate”.

Response: we have written “attenuata” instead of “attenuate” in the revised text (P14, L669).

L444: replace “the“ with “a”.

Response: we have written “a” instead of “the“ in the revised text (P14, L671).

L449: add “the” before “levels”.

Response: we have changed “to levels of” to “to the levels of” in the revised text (P14, L676).

L456: “HR-type necrosis” – please define.

Response: “HR” is the abbreviation of “hypersensitive type of resistance” (Kira´ly, et al., Journal of General Virology 2008, 89, 799–808). Please refer to the information in the list of abbreviations (P2). Some plants (such as tobacco) tend to resist the infection of pathogen by the way of localized tissue necrosis, implying that the plants have hypersensitive resistance to the pathogen. In other words, a typical HR is often characterized by localized necrotic lesions around the infection sites in plants. Therefore, the localized tissue necrosis mentioned above was called “HR-type necrosis”.

L466: you didn’t put species author names anywhere in the text.

Response: Thank you for your reminding, we have added the author name after the species name for some species in the new revised text. For example, Arabidopsis thaliana L. (P7, L307; P8, L355); Oryza sativa L. (P8, L355); Hordeum vulgare L. (P8, L356); Nicotiana benthamiana L. (P12, L528); Catharanthus roseus L. (P10, L498); Vicia faba L. (P13, L644); Solanum lycopersicum L. (P14, L664); Spinacia oleracea L. (P13, L605-606).

L474: do not put comma after “while”.

Response: we have changed “while, the activity” to “while the activity” in the revised text (P15, L710).

L474 & 475: delete space between “2,” and “3”.

Response: we have changed “2, 3-dichlorophenoxyacetic acid (2, 3-D)” to “2,3-dichlorophenoxyacetic acid (2,3-D)” in the revised text (P15, L710-711).

L478: delete “abscisic acid” and leave just the abbreviation. Misspelled “Zaetin” – it should be “zeatin” (small caps too).

Response: we have changed “downregulated by abscisic acid (ABA), Zaetin,” to “downregulated by ABA, zeatin,” in the revised text (P15, L714).

L480: “also could” -> “could also“

Response: we have changed “also could” to“could also” in the revised text (P15, L716).

L491: “react” instead of “response”.

Response: we have written “react” instead of “response” in the revised text (P15, L726).

L492: “DPI” – please define.

Response: “DPI” is the abbreviation of diphenylene iodonium in the revised text (P8, L357).

L500, 501, and so on: “cis” should stand in italics.

Response: we have changed all the “cis” to “cis” in the revised text (P15, L736, L738, L740, L744. L746, L751…).

L502 & 509: put “Arabidopsis” in italics also.

Response: we have changed all the “Arabidopsis” to “Arabidopsis” in the revised text (P15, L739, L749…).

L507: delete comma before “further” and replace “complicated” with “complex”.

Response: we have changed “, further indicate the complicated” to “further indicate the complex” in the revised text (P15, L744).

L510: “different treatments” – I would say “various life regimes” or similar.

Response: we have changed “different treatments” to “various life regimes” in the revised text (P15, L747).

L511: maybe to add “increased” before “calcium”, since calcium is not a treatment. “hormone application” too. Also reconsider “cellular development”, since is not a “treatment”.

Response: we have changed “calcium, hormones as well as cellular development” to “increased calcium, hormones application as well as cellular differentiation and growth” in the revised text (P15, L747-748).

L515: “is possible the role” – wrong syntax.

Response: we have changed “is possible the role” to “may play a role” in the revised text (P15, L752).

L529: Put “[146]” after “Yoshioka et al.” and delete from the sentence end.

Response: we have corrected these in the revised text (P16, L800). In addition, the reason that [146] become [148] are the same as above.

L531: “showed” instead of “found”.

Response: we have written “showed” instead of “found” in the revised text (P16, L702).

L543-544: “and so on” is not an adequate scientific term. You should omit it.

Response: we have changed “hormones (like ABA and ET), and so on.” to “and hormones (like ABA and ET).” in the revised text (P16, L814).

L544: “complex” instead of “complicated”.

Response: we have corrected these in the revised text (P16, L815).

L553: full stop before “Therefore”.

Response: we have changed “, Therefore,” to “. Therefore,” in the revised text (P16, L824).

L560: replace “acquired” with “obtained”.

Response: we have changed “Similar results were also acquired” to“Similar results were also obtained” in the revised text (P16, L832).

L562: “which lays a certain molecular foundation” – please rephrase to get clearer.

Response: we have changed the sentence “implying that H2O2 mediated plasma membrane Ca2+ influx perhaps is a general mechanism for maintaining intracellular Ca2+ dynamic balance, which lays a……” to “These results showed that H2O2 mediated plasma membrane Ca2+ influx perhaps is a general mechanism for maintaining intracellular Ca2+ dynamic balance, and lay a……” in the revised text (P16, L833-835).

L596: remove “the activity of”.

Response: we have changed “activate the activity of NOXs/RBOHs” to “activate NOXs/RBOHs” in the revised text (P18, L886).

L597: “in vitro”, “in vivo” should stand in italics.

Response: we have corrected all “in vitro” and “in vivo” into “in vitro” and “in vivo” in the revised text (L294, L287, L832, L887, L907, L1114).

L606: “involved” instead of “involving”.

Response: we have changed “was found involving in” to “was found involved in” in the revised text (P18, L896).

L610: add “the” before “regulation”.

Response: we have changed “in regulation of” to “in the regulation of” in the revised text (P18, L900).

L612: put semicolon after “ROS”.

Response: we have changed “ROS,” to “ROS;” in the revised text (P18, L901).

L618: delete “kinase”.

Response: we have changed “kinase activity” to “activity” in the revised text (P18, L907).

L619: “direct phosphorylation of”

Response: we have changed “direct phosphorylating” to “direct phosphorylation of” in the revised text (P18, L908).

L620: “Arabidopsis” in italics.

Response: we have changed “Arabidopsis” to “Arabidopsis” in the revised text (P18, L908).

L634: three times “and” – it should be rewritten.

Response: we have changed “in plant growth and development regulation and cellular signal recognition and transduction.” to “in plant growth control, development regulation, cellular signal recognition and transduction.” in the revised text (P18, L921-922).

L636: delete “whereas”.

Response: we have changed “whereas, receptor-like” to “receptor-like” in the revised text (P18, L924).

L656: delete comma before “directly”.

Response: we have changed “, directly binds” to “directly binds” in the revised text (P19, L982).

L660: “are also involved” and replace “with also” with “, while”.

Response: we have changed “also involve in plant immunity with also functioning” to “are also involved in plant immunity, while functioning” in the revised text (P19, L986).

L671: remove comma after “whereas”.

Response: we have changed “whereas, a LORELEI-LIKE GPI-ANCHORED PROTEIN 1 (LLG1)” to “whereas LLG1” in the revised text (P19, L997).

L676-678: “brassinolide”, “”pattern” and “botrytis” – small caps.

Response: we have changed “Brassinolide”, “”Pattern” and “Botrytis” to “brassinolide”, “pattern” and “botrytis” in the revised text (P19, L1002, L1003, L1004).

L680: “become the focus of attention” – please rephrase.

Response: we have changed “BAK1 and BIK1 both become the focus of attentions.” to “both BAK1 and BIK1 become the focus of multiple signalings.” in the revised text (P19, L1005, L1006).

L682: “recruits”.

Response: we have changed “recruit” to “recruits” in the revised text (P19, L1008).

L684: “clearer”

Response: we have changed “To be clear,” to “To be clearer,” in the revised text (P19, L1010).

L691: delete “in”.

Response: we have changed “in prior to ROS production” to “prior to ROS production” in the revised text (P19, L1017).

L702: replace “who” with “which”.

Response: we have changed “crucial factors who directly” to “crucial factors which directly” in the revised text (P20, L1040).

L703: delete “while”.

Response: we have deleted “while” from “while the activity of” in the revised text (P20, L1041).

L710: remove “in” and “LYM1 and LYM3”.

Response: we have changed “in which LYM1 and LYM3 can” to “which can” in the revised text (P20, L1048).

L714: “homolog”. In parenthesis: “…found to connect the…”.

Response: we have changed “PBL27, the Arabidopsis homologous of BIK1, was found that it connects” to “PBL27 (the Arabidopsis homolog of BIK1), was found to connect” in the revised text (P20, L1052).

L717: replace “plant immune” with “plant immunity”. “…CERK1 are organized…”

Response: we have changed “in plant immune. Whether LYK5 and CERK1 organize” to “in plant immunity. Whether LYK5 and CERK1 are organized” in the revised text (P20, L1055).

L724-725: write “…immunity, and they may also…”

Response: we have changed “…immunity, and OsRLCK185 and OsRLCK176 may also…” to “…immunity, and they may also…” in the revised text (P20, L1062).

L761: put [216] after “Duan et al.” and remove from the sentence end.

Response: we have corrected these in the revised text (P21, L1114-1115). In addition, the reason that [216] become [218] are the same as above.

L768: “serve as a two-state”

Response: we have changed “serves as two-state” to “serve as a two-state” in the revised text (P21, L1122).

L772: “FER” – please elaborate.

Response: “FER” is the abbreviation of “FERONIA”, which is a plasma membrane receptor-like kinase, is a central regulator of cell growth that integrates environmental and endogenous signals. Please refer to the information in the revised text (P21, L1126) and the list of abbreviations (P2-3).

L778: “endoplasmatic reticulum” should not be italicized nor capitalized.

Response: we have changed “Endoplasmic reticulum” to “endoplasmic reticulum” in the revised text (P21, L1132).

L814: “jasmonate” – small caps.

Response: we have changed “Jasmonate (JA)” to “JA” in the revised text (P23, L1177-1178).

L815: pleases rephrase the beginning of the sentence.

Response: we have rephrased the sentence “Once activated, the GTP-bound activated ROP/RACs recruit……” to “Once been activated, the GTP-bound ROP/RACs could recruit……” in the revised text (P24, L1184-1185).

L825: “normal plant”

Response: we have changed “plant normal” to “normal plant” in the revised text (P24, L1194).

L827: “is” instead of “was”.

Response: we have corrected this by writing “is” instead of “was” in the revised text (P24, L1196).

L828: delete “the”.

Response: we have deleted “the” from “the two PRRs,” in the revised text (P24, L1197).

L837: please rephrase the part of the sentence starting with “OsRac1”.

Response: we have changed “OsRac1 functions…..”to “OsRac1 plays a role at …….” in the revised text (P24, L1207).

L842: “…sufficient to fully activate NOXs…”

Response: we have changed “sufficient for fully activation of NOXs” to “sufficient to fully activate NOXs” in the revised text (P24, L1212).

L846: “…roles in full activation”

Response: we have changed “roles for fully activation” to “roles in fully activation” in the revised text (P24, L1216).

L849: “their N-terminal region,”

Response: we have changed “the N-terminal region of them,” to “their N-terminal region,” in the revised text (P24, L1219).

L850: “cytosol” instead of “cytosolic”.

Response: we have written “cytosol” instead of “cytosolic” in the revised text (P24, L1220).

L854: write “…in cytosol induce its efflux from the intracellular stores.”

Response: we have changed “…in cytosolic play a positive role inducing Ca2+ efflux from intracellular Ca2+ stores.” to “…in cytosol induce its efflux from the intracellular stores.” in the revised text (P24, L1224).

L858: finish the sentence after “manner” and start the next with “Furthermore,”

Response: we have changed “, furthermore,” to “. Furthermore,” in the revised text (P24, L1228).

L870: delete “stress factors” after “abiotic”.

Response: we have changed “abiotic stress factors and biotic stress factors” to “abiotic and biotic stress factors” in the revised text (P25, L1257).

L880-881: relocate “indirect” to stand before “BIK1”.

Response: we have changed “BIK1 mediated indirect way.” to “indirect BIK1 mediated way” in the revised text (P25, L1267-1268).

L887: maybe to replace “literatures” with “reports” or “studies”.

Response: we have changed “literatures” to “studies” in the revised text (P25, L1274).

L888: delete “etc”.

Response: we have deleted “, etc” from “ET, etc” in the revised text (P25, L1275).

L902: “mutant of” instead of “mutant in”.

Response: we have changed “mutant in” to “mutant of” in the revised text (P25, L1289).

L913: “complexes which act as”

Response: we have changed “complexes so that acts as” to “complexes which act as” in the revised text (P25, L1299-1300).

L922: “α-subunit” instead of “a-subunit”.

Response: we have changed “a-subunit” to “α-subunit” in the revised text (P26, L1319).

L927: add “production” after “H2O2”.

Response: we have added “production” after “H2O2” in the revised text (P26, L1324).

L928: delete “for H2O2 production”.

Response: we have changed “regulates NOX/RBOH for H2O2 production” to “regulates NOX/RBOH” in the revised text (P26, L1325).

L938: “involved” instead of “involving”.

Response: we have changed “involving in the regulation” to “involved in the regulation” in the revised text (P26, L1334).

L950: delete comma after “that”.

Response: we have deleted “,” from “that,” in the revised text. (P26, L1347).

L955: delete “Where,”

Response: we have deleted “Where,” in the revised text (P26, L1352).

L956: add “being” after “not”.

Response: we have changed “seems not involved” to “seems not being involved” in the revised text (P26, L1353).

L960: should “pathway” stand at the end of the sentence?

Response: we have changed the phrase “via the JA/ET and/or SA pathways” to “via the pathways of JA/ET and/or SA” in the revised text (P26, L1355).

L964: put “highlight” instead of “emphasize”.

Response: we have changed the word “emphasize” to “highlight” in the revised text (P26, L1361).

L973: replace full stop with comma and continue the sentence.

Response: we have changed the phase “. While, mutation” to “, While mutation” in the revised text (P27, L1380).

L976: “earlier”?

Response: we have changed the phase “it has found early” to “it has found earlier” in the revised text (P27, L1382-1383).

L982: strange font size for “I”.

Response: we have corrected font size of “I” in the revised text (P27, L1389).

L985: “development of the studies on the”

Response: we have changed the phase “developing of the studies in the” to “development of the studies on the” in the revised text (P27, L1392).

L900-991: needs to be rephrased.

Response: we have rephrased all these in the revised text. Please refer to the detailed information from L1273 to L1414. Most of all, we rephrased the sentence “Currently, NADPH oxidases are emerging as the focus of researches, for being the roles of ROS producer, multiple signaling centers, and diverse function implementers in plants.” to “Currently, NADPH oxidases, being the ROS producers, multiple signaling centers and diverse function implementers in plants, are becoming the focus of studies.”in the revised text (P27, L1397-1398).

L993: “complex” instead of “complicated”.

Response: we have changed the word “complicated” to “complex” in the revised text (P27, L1399).

L994: add “germination” after “pollen”.

Response: we have changed the phase “in pollen” to “in pollen gemination” in the revised text (P27, L1401).

L1000: “to be answered in the future, for instance...”

Response: we have changed the phase “to be answered further. For instance…” to “to be answered in the future, for instance...” in the revised text (P27, L1407).

L1005: “have been obtained”; L1006: “items exhibiting”; L1007: add “while only” before “few”

Response (to L1005, L1006, L1007): we have rephrased the sentence “Although big achievements have obtained on NOXs/RBOHs and their related items, which exhibit huge potentials in improvement for crop yield and quality, almost all the results obtained now are just on the basis of model plants, few field crops have been considered for deep research.” To “However, almost all the results obtained now are just on the basis of model plants, only few field crops have been considered for deep research.”. Please refer to the information in the revised text (P27, L1411-1413).